# Quantifying the Evolution of Giant Panda Habitats in Sichuan Province under Different Scenarios

**Zhenjiang Song [1], Baoshu Wu [2], Wenguang Xiong [3], Lan Gao [4],\* and Yi Li [4],\***

1    Institute of New Rural Development, College of Economics and Management, Jiangxi Agricultural University, Nanchang 330045, China
2    School of Business Administration, Jiangxi University of Finance and Economics, Nanchang 330032, China
3    School of Foreign Languages, Jiangxi Agricultural University, Nanchang 330045, China
4    College of Economics and Management, South China Agricultural University, Guangzhou 510642, China
\*    Correspondence: gaolan@scau.edu.cn (L.G.); xiaoyi1983524@126.com (Y.L.); Tel.: +86-137-2484-9061 (L.G.); +86-139-2228-0524 (Y.L.)

**Abstract:** The giant panda (*Ailuropoda melanoleuca*) is a relic species in China and a flagship species in the field of endangered wildlife conservation. The conservation of the giant panda's habitat has gained widespread attention for this reason. Historically, Chinese Giant Panda Nature Reserves are surrounded by communities and the resource utilization behavior of households disturbs the giant panda habitat. Changes in these communities and in Giant Panda Nature Reserves began around 2010, with the feminization and aging of the farm labor force. These changes brought different resource utilization behaviors that led to different evolutionary tendencies in giant panda habitats. This research study assesses the impact of these tendencies based on data from the fourth survey of the giant panda in the Sichuan Province and from geographic information data. The paper aims to uncover the internal mechanisms of farmers' resource utilization behavior in terms of the changes wrought to giant panda habitats. The paper simulates the future habitat of the giant panda based on the LUCC (land use/cover change) model to identify anticipated changes in future landscape patterns and the habitat quality of giant pandas under the current scenarios. The paper analyzes the spatial-temporal change of landscape patterns through the land use transfer matrix, based on the Markov model. The results of the scenario analyses illustrate the spatial and temporal difference in habitat quality. The driving mechanism for landscape pattern change is explored using the logistic regression model. The paper simulates the variation tendency of giant panda habitats under differential labor force structures and resource utilization behavior based on the CA (cellular automata) model, with the robustness of the results verified by participatory experiment. Through four scenarios of simulated farm labor force structure and resource utilization behavior, results suggest that the quality of giant panda habitats in the future will be relatively high when workforce feminization and aging is intense and resource utilization behavior is weakened. The simulated results confirm that the current agricultural labor force structure can reduce the intensity of resource utilization behavior. In this scenario, disturbance to giant panda habitats would decrease and their quality would improve.

**Keywords:** structure of the labor force; resource utilization behavior; giant panda habitat; LUCC; landscape pattern

## 1. Introduction

The giant panda (*Ailuropoda melanoleuca*) is a special relic species in China and a flagship species in the field of endangered wildlife conservation. Giant panda habitat conservation has, therefore, gained widespread attention [1]. For historical reasons, the Chinese Giant Panda Nature Reserves are surrounded by communities, and human resource utilization activities continue to affect the land use pattern of nature reserves, their surrounding habitats, and their potential habitats. These activities lead to habitat fragmentation [2–4],

corridor fracture [5], and population isolation of giant pandas [6–11] and are not conducive to the sustainable development of the species [12,13]. Over the past few centuries, human hunting and encroachment have caused a sharp decline in the number of giant pandas [14]. Further, the particularity of the panda's physiological characteristics and reproductive modes have accelerated shrinkage in the giant panda population, causing the species to become endangered, which their fragile habitats exacerbate [15,16]. In recent years, even though the habitats remain fragile, both the habitats and corridors have been effectively protected and expanded [17].

Nature reserves and their surrounding communities are special functional ecological areas that aim to protect rare and endangered species. Activities that threaten the protection of flagship species are regarded as disturbance behaviors [18,19]. As the range of farmers' resource use activities intersects with the habitat of flagship species, it is regarded as typical disturbance behavior [19,20]. Regulations on the management of nature reserves stipulate that felling, grazing, hunting, fishing, medicinal plant collection, reclamation, burning, mining, quarrying, sand digging, and other activities in nature reserves are prohibited. Further, many non-governmental organizations (NGO), the national Nature Reserve Administration, and local governments carry out development projects, such as community co-management and construction of energy-saving facilities in nature reserves and their surrounding communities [21]. These measures have not stopped farmers from relying on natural resources, and the root of this phenomenon has been identified as farmers' inertia. Current agricultural resource utilization activities are part of traditional economic activities [22], meaning that farmers have a high dependence on gathering firewood, cultivating land, and exploiting other resources, although establishing the nature reserves has restricted these traditional activities. However, the traditional use of such resources has not changed fundamentally even with protective measures, and the interference caused has existed for a long time [22,23].

The human population of the Giant Panda Nature Reserve and its surrounding communities is as high as 14.159 million. This includes a rural population of 9.894 million, which imposes resource pressures on the nature reserve. Farmers' behaviors are closely related to their livelihoods, including planting (for self-sufficiency), firewood use (for cooking, smoking foods, and heating), and other interference behaviors [21,24,25] and are still widespread in nature reserves and surrounding communities. The contradiction between protection and development remains acute, and the trend of production in urban and rural areas is irreversible. Men, middle-aged workers, and young people from the nature reserves and surrounding communities migrate to the cities where they can earn higher incomes [26,27] and acquire labor literacy [28,29]. This trend is leading to the feminization and aging of the remaining labor force [30,31]. However, the constraints of the natural geographical environment are severe: the slope is over 25° on the cultivated land, fragmentation is serious, and it is difficult to carry out mechanical tillage. The mountains are steep for firewood cutting, the forests are dense, and aggressive wild animals, such as wild boar and black bear, often appear [32,33]. In such circumstances, the intensity of resource use is reduced and the structure of livelihood resource use is adjusted, thereby decreasing disturbance to the regional ecological landscape pattern and diminishing the impact on giant panda habitats [34,35]. With the trend of feminization and aging in the farm labor force and changing resource utilization behaviors in farmers' households, interference with giant panda habitats is, nonetheless, still occurring, and a variety of scenarios are needed to assess future changes to giant panda habitats.

In terms of research approaches, the major methods of habitat simulation are CA (cellular automata)-Markov, IMAGE, GTR, CLUE-S, and so on. CA-Markov is used to study the simulation of land utilization with multi-time scales and relatively higher precision than other models. The model has been used to simulate land-use change in small ranges, such as nature reserves [36–38]. IMAGE is used to simulate the effects of environmental factors of land use/cover change (LUCC), making an important contribution to building the Human-Climate-Biosphere System model. This model is used for large scale research,

such as changes in land use and landscape patterns in South China [39], and the analysis of LUCC and landscape fragmentation on a large scale [40,41]. GTR has improved the Von Thunen model, used to study the change in land utilization on a large scale. Long et al. (2001) and Ge et al. (2018) analyzed the LUCC in the Yangtze River basin and cultivated land use patterns in China based on GTR [42,43]. CLUE-S is used to analyze and quantify the relationship between LUCC and social-ecological systems to simulate the evolution of land utilization. The model is more applicable to LUCC simulation [44,45].

This paper explores the changing trajectory of future giant panda habitats. CA-Markov can describe multi-stage habitat evolution, whose model is based on LUCC. CA-Markov is more relevant than other models, in relation to spatial habitat pattern change simulation on a small scale, such as nature reserves, indicating that it is applicable to studying the internal characteristics of habitat change in the study area, thus, providing a fit for analyzing the spatial and temporal patterns of habitat change. Therefore, the paper uses CA-Markov to simulate the long-term change to giant panda habitats. However, the available literature on landscape spatial patterning is drawn from physical geography and social economy. Therefore, the research verifies the robustness of the CA-Markov results by participatory experiment only to confirm the future direction of giant panda habitat change. The data for this study comes from investigations of the research group in the Wawushan Nature Reserve, Sichuan Province, collected in July 2019. The research aims to provide a reference point for establishing a national park system and formulating future giant panda conservation and management policies.

## 2. Materials and Methods

### 2.1. Study Area

The simulation experiment of giant panda habitat change was conducted in the Wawushan (WWS) Provincial Nature Reserve Area located in Hongya County, Meishan City, Sichuan Province, between WWS Town and Gaomiao Town or between latitude 29°25′–29°34′ north and longitude 102°49′–103°00′ east. The area covers 36,490 hectares, with 26,498.3 hectares in the core area, 5826.3 hectares in the buffer area, and 4165.5 hectares in the experimental area. The WWS Nature Reserve is located in the Daxiangling Mountain System, where the population structure of the giant panda is concentrated (only three local populations), and has typical characteristics of habitat structure integrity. Further, no community distribution is evident in the WWS Nature Reserve and there are few outside communities, meaning that its typicality is easier to analyze (Figure 1).

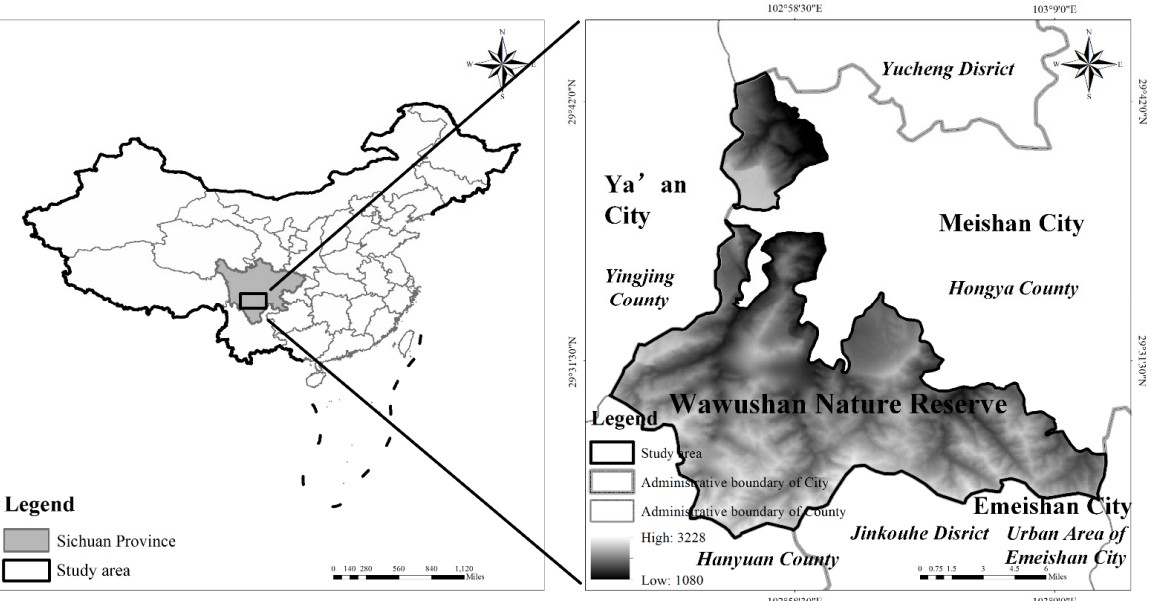

**Figure 1.** Study area.

### 2.2. Data Source and Pre-Processing

2.2.1. Landscape Pattern Classification

The image data adopted in this paper is from 2015 and 2017, derived from the geospatial data cloud open data Landsat-8 OLI, with a spatial resolution of 30 m. The images were taken between June and August when there is less cloud cover, rich vegetation, and easy recognition of surface features. To reduce the difficulty of atmospheric correction, images were filtered with less cloud.

The image band synthesis, geometric correction, radiometric calibration, atmospheric correction, and cloud removal were processed in ENVI5.0. As the study area (the WWW Nature Reserve) spans four maps, image stitching was required. The remote sensing image of the study area was obtained by image clipping based on the boundary file and the one kilometer buffer file of the Giant Panda Nature Reserve. The boundary files and the buffer file were used to carry out image cropping, resulting in the study area displayed in remote sensing image. The supervised classification scheme adopted in this paper is based on the Land Cover Classification System of Environmental Remote Sensing Monitoring in the Decade of Ecology (from the National Remote Sensing Center of China), which divides the landscape pattern of the Giant Panda Nature Reserve and its external buffer zone into eight first-level categories and sixteen second-level categories. This paper interpreted and extracted the economic management category and pattern information of the study region in 2015 and 2017 by combining supervised classification and visual interpretation. Meanwhile, for disputed ground object information, the survey samples from July 2018 to May 2019 and the existing atlas were used for comparison and correction, ensuring that the accuracy of first-level and second-level ground objects reached 90% and 84%, respectively, to meet the current research needs.

2.2.2. Driving Factors

Natural factors and human activities affect the habitat of giant pandas. The natural factors driving habitat change include the digital elevation model (DEM), slope, aspect, and giant panda activity traces (The discovery point of excrement could verify that giant pandas used the habitat here. Therefore, the discovery point of excrement could represent the giant panda activity trace, which is an important basis for habitat delineation). Human disturbance includes general disturbance (logging, bamboo cutting, planting bamboo shoots, grazing, crop planting, wood cutting, hunting, road activity, etc.) and large-scale disturbance (hydro-power stations and reservoirs, high-grade roads, transmission lines, mining, scenic spots, etc.). These selected factors rely on the survey reports of the giant panda. Remote sensing data came from the Geospatial Data Cloud, such as DEM, slope, and aspect. Basic geographic information came from the National Geomatics Center of China, including details of roads, hydro-power stations and reservoirs, transmission lines, mining activity, and scenic spots. Habitat data came from survey reports on the giant panda, such as giant panda activity traces, logging, bamboo cutting, bamboo shoots planting, grazing, and hunting. Data about planting and wood cutting came from field investigations using 64 questionnaires, where no more than 200 households existed, according to the WWW Nature Reserve authority.

In this study, farmers' resource use behavior (planting and firewood use) with dominant characteristics in the study area were selected as the mediation variables for the impact of the farmers' labor structure on giant panda habitat change. This decision is supportable. There is generally no significant disturbance in the study area (11 nature reserves). (The area comprises protected spaces mostly without human habitation; high-grade highways mostly adopt tunnel mode; large scenic spots are generally outside the protected area; mines have been closed; hydro-power stations have been shut down on a large scale; etc.) Some small disturbance activities have been effectively controlled (i.e., strict cutting-quota control, planned bamboo-shoot planting, and forbidden commercial bamboo-shoot planting; a few minority communities (Tibetan, Yi) carry out grazing; forest roads have been abandoned (the WWW Nature Reserve is typical); poaching and hunting is strictly prohib-

ited). However, farmers' planting and fuelwood utilization behaviors continue because farmers' livelihoods are dependent on them. This paper selected household resource use behavior (planting and fuelwood use behaviors) and other human disturbance factors, such as roads, to assess the evolution of giant panda habitats.

The digital elevation model (DEM) adopted in this paper is ASTER GDEM002 with a spatial resolution of 30 m. The elevation data of the study area were obtained by projection transformation and clipping. Slope and aspect data based on DEM images were achieved using the spatial analysis tool in ArcGIS 10.1. The data of giant panda activity traces came from the fourth giant panda survey report of Sichuan Province. Land cover data were derived from the foregoing remote sensing interpretation. Farmers' planting and fuelwood use behavior in the analysis were based on the survey data of resource use location in the questionnaire. Road data come from road vector data published by the National Geographic Information Center.

### 2.3. Method

The framework for this part includes: analysis of spatio-temporal pattern change of landscape patterns, analysis of habitat quality, analysis of driving forces of landscape patterns, simulation of habitat quality, verification of simulation results. The analysis of driving forces of landscape patterns was based on the logistic regression model. The simulation of habitat quality was based on the CA-Markov model. The verification of simulation results was based on participatory scenario method.

#### 2.3.1. CA-Markov Model

1. The Markov Model is widely used in the research of random motion processes [46]. It determines the trend of landscape pattern change over time by measuring the initial probability of different states and the transition probability relationship between states to achieve landscape pattern change prediction [47]. The change of landscape type is usually anisotropic, that is, the land type can change from the current type to other types, or from other types to the current type. During the transition process, the types and number of giant panda are constantly changing. The key problem that the Markov Model solves is to determine the transition probability among different landscapes in complex landscape pattern change. The essence of landscape pattern change is that each transition probability constitutes a transition probability matrix:

$$P = \begin{bmatrix} P_{11} & \cdots & P_{1n} \\ \vdots & \ddots & \vdots \\ P_{n1} & \cdots & P_{nn} \end{bmatrix} \tag{1}$$

where, $P_{ij}$ represents the transition probability of landscape type I to landscape type J, and $P_{ij}$ should meet two conditions: $0 \leq P_{ij} \leq 1$ and $\sum_{j=1}^{n} P_{ij} = 1$. N is the number of landscape types [47].

2. Cellular automata (CA) is a discontinuous space-time dynamic model characterized by discrete time, space, and state [47–50]. The CA system consists of the cellular element and its state, cellular space, cellular neighborhood, and transformation rules. Each cell in cellular space has a finite number of specific states, updated synchronously according to multi-defined partial rules. These partial rules interact with each other to form a dynamic evolutionary system. The general expression of CA is:

$$C_{(t+1)} = f(C_{(t)}, N) \tag{2}$$

where C is finite and has a discrete set of states of cellular. t and t + 1 are different moments. N is the neighborhood of cellular. f is the cellular transformation rule of local space [47].

3. The CA-Markov Model is supported by the GIS platform that determines the transition of the cellular state by the transforming area matrix and conditional probability image

operations to simulate the change of landscape pattern [47,51]. The specific operation process is as follows:

- Superposition analysis: By superposition analysis, landscape type transition probability matrix, transition area matrix, and a series of conditional probability images are obtained. These images come from the transition probability matrix, representing the probability that each pixel will be covered by a landscape type at the next moment.
- Construct CA filter: According to the distance between neighbor and cell, the weight factor with significant spatial significance is created to make it act on the cell, so as to determine the state change of the cell. In this study, a $5 \times 5$ filter was used; that is, a rectangular space consisting of $5 \times 5$ cells around a cell was considered to have a significant influence on the change of the cell state.
- Determine the start time and the number of CA cycles: Based on the landscape type transition matrix of the study area from 2000 to 2017, the year 2017 was taken as the starting point for landscape pattern prediction, and the iteration number of the CA model was eight. The landscape pattern of the Giant Panda Nature Reserve and its external buffer zone in 2025 was simulated under the natural growth scenario.

### 2.3.2. Logistic Regression

At present, logistic regression (LR) is widely used to study the driving forces of land use and landscape change [52–55]. According to the current state of the study area, the driving forces of landscape pattern change were studied by LR with human and natural factors as independent variables. The general expression of LR is:

$$Y = lg[P_i/(1 - P_i)] = \alpha + \beta_1 X_1 + \beta_2 X_2 + \ldots + \beta_n X_n. \qquad (3)$$

where $P_i$ is the probability that a certain landscape type may appear in each grid. $X_n$ represents each influence factor, $\alpha$ represents a constant term, $\beta_1, \beta_2 \ldots, \beta_n$ is the partial regression coefficient of LR, and $X_n$ represents the influence of the independent variable on Y. When the Y value is positive (negative) and statistically significant, the occurrence of Y increases (or decreases) with the increase in the corresponding independent variable value under the condition that other independent variables are controlled [47,56].

### 2.3.3. Giant Panda Habitat Change Scenario Setting

The participatory scenario took the form of a workshop. Its aim was to explore what would occur in the future, when farmers faced challenges. The procedure considered five factors: background description of the study area, stakeholder description, exploratory scenario design, foreseeability scenario design, and result analysis. The principle of the experiment was to retain the traces of households.

Due to uncertainty about the future labor structure and resource utilization intensity of farmers, the simulation of giant panda habitats must consider multiple trends to approach the real situation. Therefore, simulation of the giant panda habitat needs to consider a variety of trends approximating the real situation. Based on the regional land use transfer matrix from 2015 to 2017, different scenarios of future habitat evolution were designed for the WWW Nature Reserve by controlling the probability of the land use transfer matrix and combining it with natural and human factors that affect regional habitat change. Thus, the size and structure of giant panda habitat landscape types under different farmer–labor structure and resource utilization intensity scenarios were obtained. Finally, the CA-Markov model was used to simulate spatial layout of habitats in different scenarios. The habitat evolution of giant pandas in the WWW Nature Reserve from 2017 to 2025 is as follows:

- Scenario 1: Ecological priority (with the highest degree of feminizing and aging of the peasant labor force)

Presently, the national park covers the former WWW reserve and its surrounding communities, which have banned planting. All cultivated land has been converted to forest. Part of the collective forest is included in the national park, which is off limits to humans and farmers' resource use. Most of the collective forests in the surrounding communities that are not included in the national parks have been included in ecological public welfare forests. These forests forbid timber and firewood cutting, which can be renewed after 35 years. Firewood logging is banned from the few remaining commercial forests and only modest regeneration is allowed at minimal quotas. In this scenario, the government replaces the ecological migration policy by guiding farmers to transfer employment, as such policies are costly and divisive. The feminizing and aging of the rural household labor force in regional communities increase, and the community population has negative growth. Farmers buy food from the market and use clean energy, such as biogas and solar energy, as fuel (communities, villages, or individuals who build new biogas digesters or buy new solar energy can receive subsidies). Their dependence on natural resources (arable land, woodland) decreases. In this scenario, the probability of farmland transferring to forest land, and construction land and unused land transferring to forest land is increased by 100%.

- Scenario 2: Coordinated development (with feminization and aging of the rural labor force slowing)

In this scenario, the region is planted with characteristic crops mainly (white tea, Yalian, etc.), supplemented by food crops. As the planting scale cannot be increased, some farmers have joined a new round of projects to return farmland to forest. In terms of forest land use, collective forests are not included in national parks, and selective cutting for tending purposes is allowed in ecological public welfare forests. The target of cutting can meet the demand of farmers for firewood. At the same time, alternative energy (biogas, solar energy) projects in the community are steadily advancing. Biogas and other clean energy has occupied a dominant position in farmers' household energy systems. In terms of the labor structure, the feminizing and aging trend of the rural household labor force is slowing. Local governments and protected areas support farmers in ecotourism. Ecological jobs (forest rangers, etc.) become available to villagers on a voluntary basis with reduced requirements (in terms of age, gender, education level, etc.). Part of the male and young adult labor force in the village is harnessed for employment. In this scenario, the probability is that 25% of cultivated land and 50% of construction land and unused land will transfer to forest land.

- Scenario 3: Partial constraint (with a high degree of feminizing and aging of the labor force)

In the partial-constraint scenario, the scale of the characteristic planting industry is limited and a new round of farmland conversion projects is effectively promoted. In woodland utilization, part of the collective forest is delimited and incorporated into the national park. This part of the forest is off limits to human access and farmers' use of resources is prohibited. The ecological public welfare forest that has not been delimited prohibits felling. It is difficult to obtain firewood and the commodity forest cutting index is low. With the help of local governments and protected areas, biogas and solar projects have made clean energy the main energy source for rural households. In terms of the labor structure, additional ecotourism activities are forbidden because they disturb the habitat of giant pandas. Established farmhouses are tightly regulated; ecological employment positions have higher requirements (e.g., age, gender, physical ability) for which few farmers are qualified. Regional farmers still migrate out for work as their main livelihood strategy. The feminizing and aging of the rural labor force are high. In this scenario, the probability is that 50% of farmland and 100% of construction land and unused land will transfer to forest land.

- Scenario 4: Current situation (with a high degree of feminizing and aging of the labor force)

Under the current situation, characteristic bamboo shoot and grain planting without restriction dominate the planting industry, and farmland conversion projects have not been implemented. In terms of forest land utilization, the scope of national parks does not exceed the protected areas and collective forests are not included in national parks. Although there are few targeted areas for commercial forest harvesting, thinning and firewood harvesting are allowed. In terms of labor force structure, ecotourism operations are allowed in experimental zones of the protected areas. New farmland in other areas and the expansion of farming is prohibited. Temporary employment of rangers is allowed. Men and young adults either work in nearby market towns (Wawushan Town and Liujiang Town) or transfer to Meishan, Chengdu, and the southeast coastal areas for employment. The feminizing and aging of the rural household labor force in the region are relatively high.

Based on the probability matrix of land use transfer from 2015 to 2017 and the proportion of each land use type in 2017, this study adopts the Markov model to simulate the proportion of each land use type in 2025 under natural development scenarios (as shown in Table 1).

**Table 1.** Land use structure, Wawushan Nature Reserve, under four scenarios in 2025 (%).

| | | 2017 | Ecological Priority | Coordinated Development | Partial Constraint | Current Situation |
|---|---|---|---|---|---|---|
| Farmland | | 7.96 | 0 | 1.44 | 0.55 | 8.06 |
| Woodland | | 74.20 | 87.77 | 81.50 | 83.03 | 72.27 |
| Grassland | | 15.09 | 10.45 | 15.09 | 14.58 | 14.53 |
| Waters | | 1.97 | 1.78 | 1.97 | 1.84 | 2.05 |
| Construction land | | 0.72 | 0 | 0 | 0 | 1.55 |
| Unutilized land | | 0.06 | 0 | 0 | 0 | 1.54 |
| Constraint of human activity [①] | Surrounding village | √ | ○ | √ | √ | √ |
| | Experimental zone | × | × | √ | ○ | × |
| | Buffer zone | × | × | × | × | × |
| | Core zone | × | × | × | × | × |
| Additional regulatory regime from government decisions [②] | | ○ | √ | × | ○ | ○ |
| Change of labor force structure [③] | Feminization of farmer labor force | √ | √ | × | ○ | √ |
| | Aging of farmer labor force | √ | √ | × | ○ | √ |

Notes: [①] √: no constraints. ○: somewhat constrained. ×: constrained. [②] √: an addition; ○: uncertain; ×:no addition. [③] √: heighten; ○: uncertain; ×: weakened.

## 3. Results and Analysis

### 3.1. Temporal and Spatial Changes of Landscape Pattern

This section considers the degree of change of landscape pattern at different time points (2015 and 2017). According to the spatial distribution of landscape types in different years and the change of first-level landscape areas in the study zone, the rate of spatial and temporal change in landscape pattern and landscape type transformations were obtained. Thus, the impact of human resource utilization on the ecological landscape pattern is revealed.

Woodland, shrub woodland, and grassland are the main landscape types in the Giant Panda Nature Reserve. The area accounts for about 99% of the total area of the study zone, of which forest land accounts for about 98% (see Figure 2). This indicates that the giant panda habitat landscape in the region is broad and the giant panda habitat structure is relatively complete. Further, from 2015 to 2017, the small hydropower stations in Wawushan were shut down. These key node events in this region caused changes in the regional landscape structure. In the study area, 66.67% of construction land was reduced due to demolition of power plants and staff dormitories, and 29.83% of the grassland, 27.75% of the wetland, and 11.76% of shrubland were reduced due to the return of farmers to planting and firewood utilization once free electric power from the power plant was unavailable. As seen from the figure to the left of Figure 3, land use changes in the core area and buffer zone occur mainly near the border of the protected area, while land use changes in the experimental area occur mostly in scenic spots. As seen in the figure on the right below, construction land has the highest turnover rate in the region, followed by

grassland, wetland, and shrubland. The area of cultivated land decreased slightly, while the areas of water and forest land increased slightly.

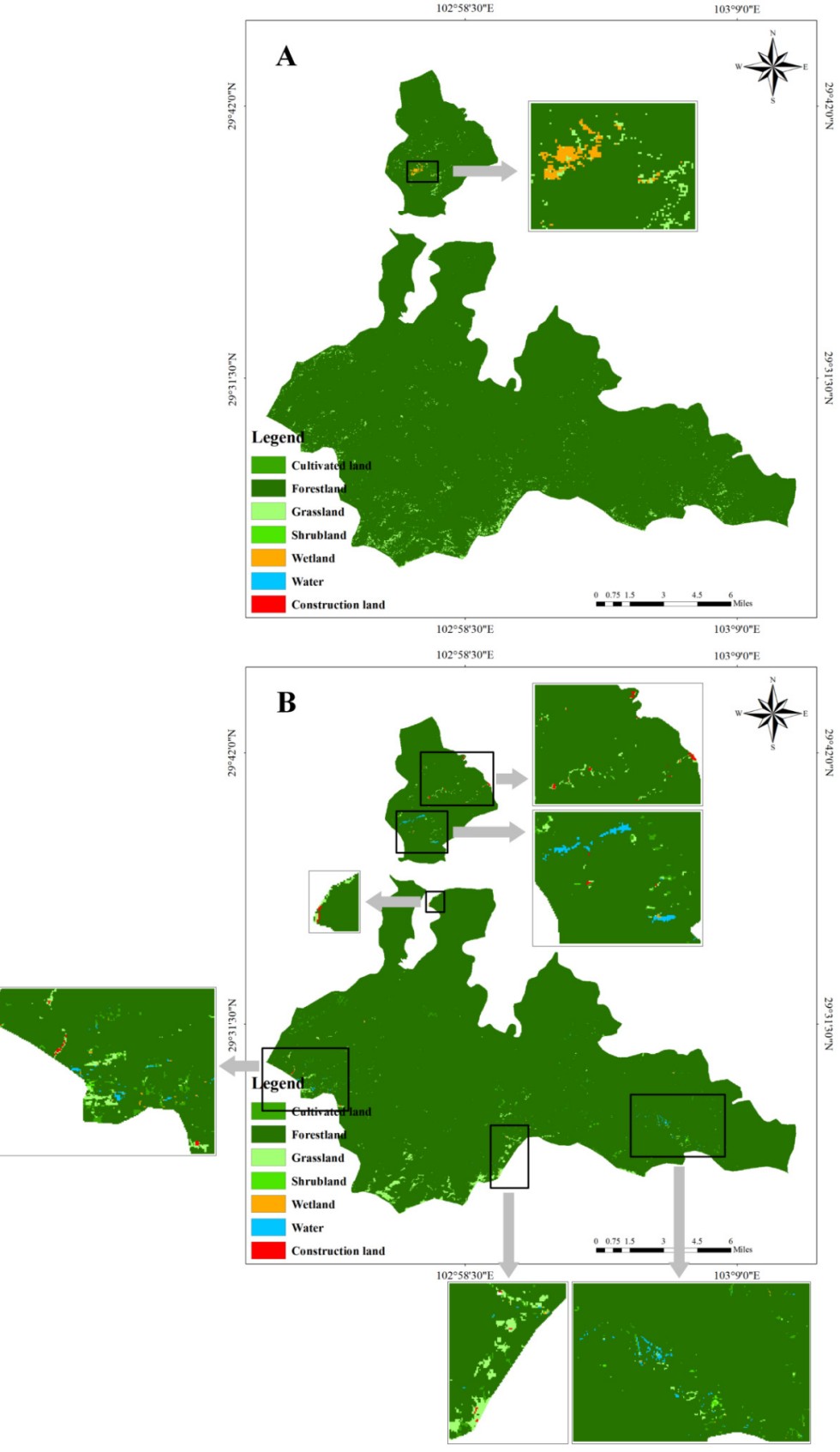

**Figure 2.** LUCC of Wawushan Nature Reserve: (**A**) 2015; (**B**) 2017.

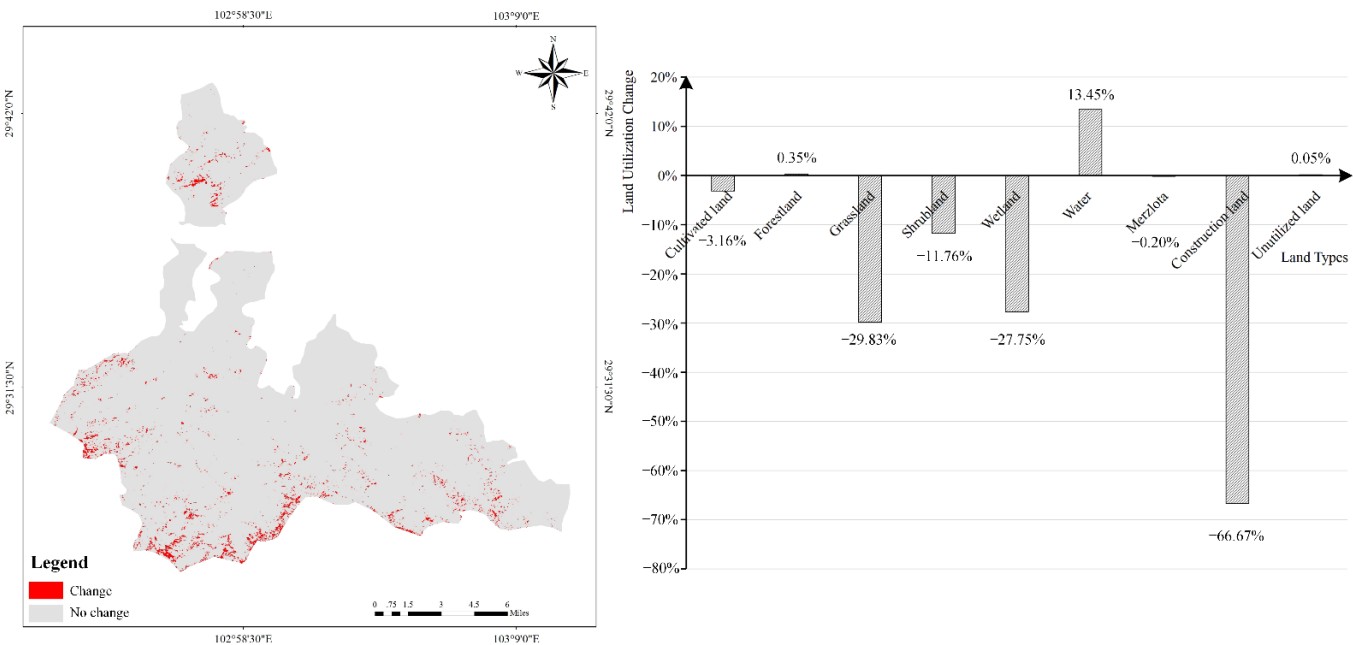

**Figure 3.** Land use change in the Wawushan Nature Reserve.

### 3.2. Habitat Quality Change

The habitat quality level can reflect landscape biodiversity and the survival suitability of flagship species. Natural breaks in ArcGIS were used to grade the output habitat quality values into five levels (Table 2) to analyze spatio-temporal differences in habitat quality.

**Table 2.** Grades of habitat quality.

| Level | Value Range | Description |
|---|---|---|
| LL | 0~0.2 | Lowest habitat quality |
| LH | 0.2~0.6 | Lower habitat quality |
| M | 0.6~0.7 | Moderate habitat quality |
| HL | 0.7~0.9 | Higher habitat quality |
| HH | 0.9~1 | Highest habitat quality |

Forest land is the core of the giant panda habitat structure, meaning its quality is high. Grassland, shrubland, and wetland are less disturbed by human activities, further indicating that habitat quality is higher. Middle-level regions were scattered throughout the study area. Low-grade areas are concentrated in the Wawushan Scenic Area, and the border area of the reserve experience frequent human activity. The corresponding landscape types are construction land and unused land (as shown in Figure 4).

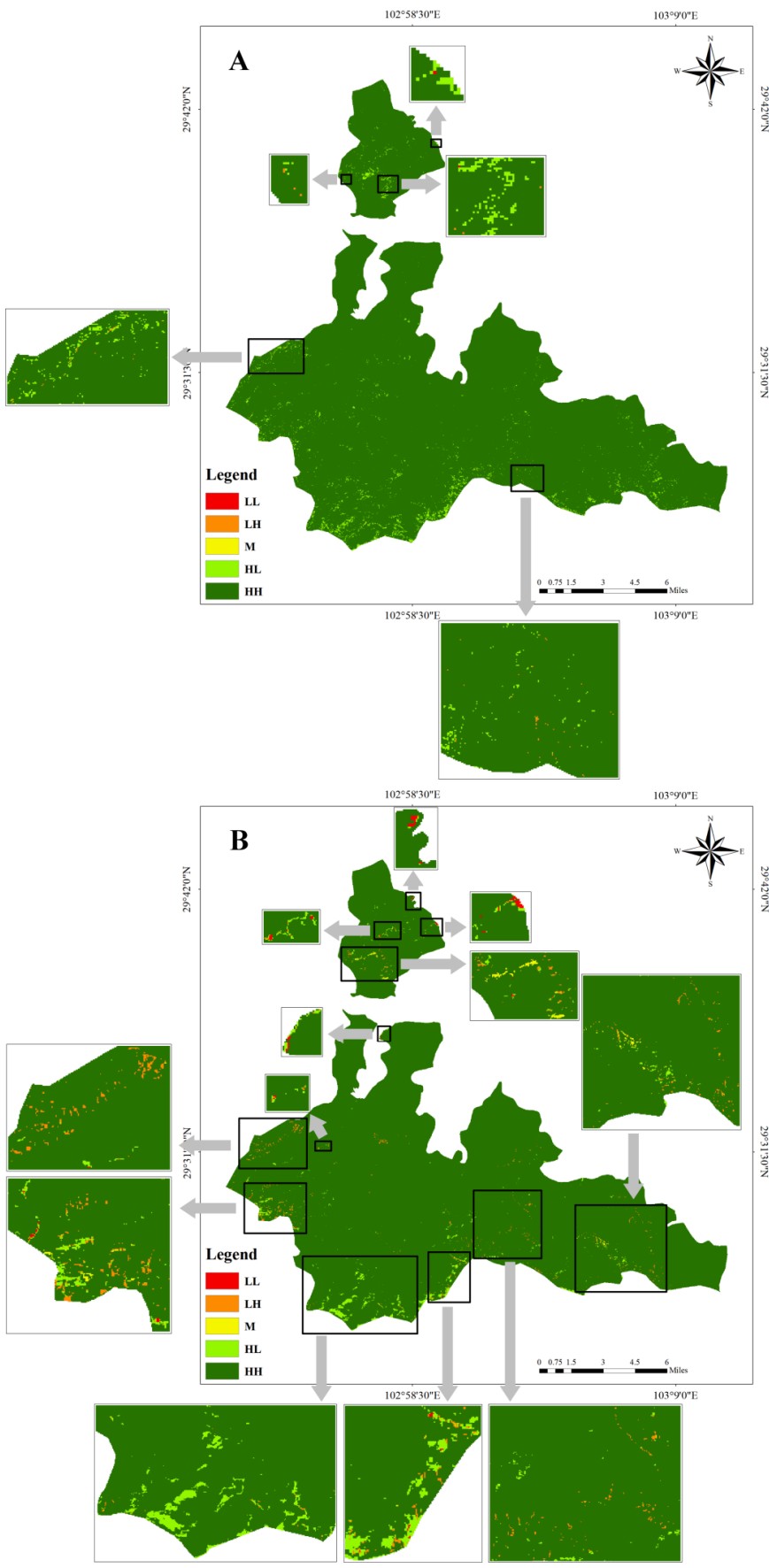

**Figure 4.** *Cont.*

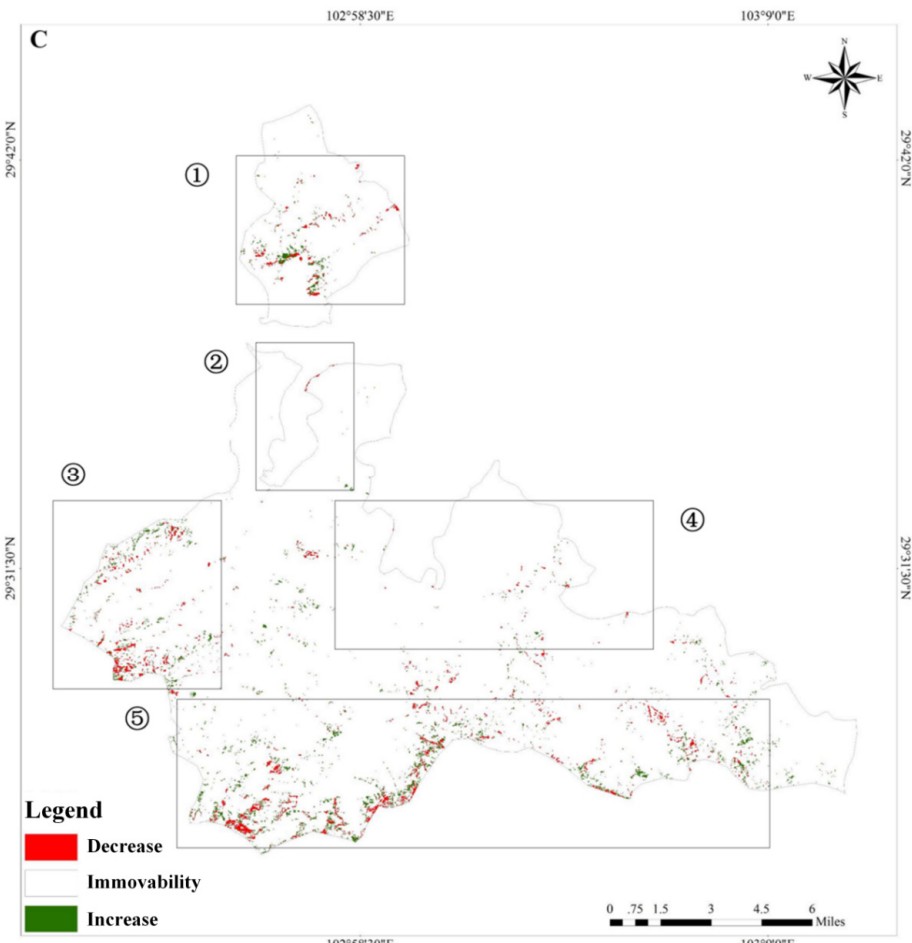

**Figure 4.** Habitat quality of the Wawushan Nature Reserve: (**A**) habitat quality of WWW nature reserve in 2015; (**B**) habitat quality of WWW Nature Reserve in 2017; (**C**) changes to habitat quality in WWW Nature Reserve from 2015 to 2017. Notes: In (**C**), section ① is the experimental area of the WWW Nature Reserve. The WWW Scenic spot is located in the area. In recent years, new service facilities and an increased in the number of tourists have led to deterioration of the quality of giant panda habitats in the area. In sections ② and ④, human resource behavior in the areas adjacent to Changhe Village, Yanyuan Village, and Heishan Village caused habitat degradation. Habitat quality declined on the border of the reserve because of a road crossing at ③. However, the habitat quality of section ③ has increased in recent years because the method of road construction combines bridges and tunnels, thereby reducing disturbance from vehicular traffic of giant panda habitats. Section ⑤ is at the southern-most part of the protected area. The quality of the habitats decreased due to the dual impact of human resource utilization activities and the climate ecotone. (**A**,**B**) both indicate that the overall quality of the giant panda habitats in the WWW Nature Reserve in the two phases is high, but there is still a trend of deterioration in local habitat quality (**C**). This is manifested in areas with frequent human activity (such as the WWW scenic spots) and on forest margins. In 2015 and 2017, the overall habitat quality score of the study area was 0.9987 and 0.9937. The area of medium grade and above decreased, and the area of low grade increased. In 2015, the proportion of habitat area of higher grade and above was 99.9695%. By 2017, the area of higher grade and above accounted for 99.4650%, a decrease of 0.5046% (as shown in Table 3). (**C**) shows that during 2015–2017, landscape types of higher quality were transformed into lower quality in some parts of the region. The change of landscape type of higher quality grade was characterized by a decrease in the vegetation index of shrubby land, and part of the original shrubby land was transformed into arable land. At the same time, the wetland area is shrinking and grassland is being degraded. In their place, arable land and unused land are expanding, which leads to the expansion of human disturbance activities and the deterioration of the overall habitat quality of the region.

**Table 3.** Area and percentage of habitat quality at all levels in the WWW Nature Reserve.

| Habitat Quality Level | Area | | | | Change in Area | |
|---|---|---|---|---|---|---|
| | **2015** | | **2017** | | **2015–2017** | |
| | **km$^2$** | **%** | **km$^2$** | **%** | **km$^2$** | **%** |
| LL (Lowest habitat quality) | 0.0018 | 0.0005 | 0.1237 | 0.0339 | 0.1219 | 0.0334 |
| LH (Lower habitat quality) | 0.1046 | 0.0287 | 1.4974 | 0.4103 | 1.3927 | 0.3817 |
| M (Middle habitat quality) | 0.0049 | 0.0013 | 0.3311 | 0.0907 | 0.3262 | 0.0894 |
| HL (Higher habitat quality) | 5.9789 | 1.6385 | 3.6883 | 1.0108 | −2.2906 | −0.6277 |
| HH (Highest habitat quality) | 358.8097 | 98.3310 | 359.2595 | 98.4542 | 0.4499 | 0.1233 |

*3.3. Analysis of Driving Forces in Landscape Pattern Evolution*

With the development of the social economy and continual improvement of communication among population groups, the unbridled use of natural resources and the impact of economic interests will inevitably change the natural ecosystem [57,58]. This will result in a decline in habitat quality and biodiversity. The landscape pattern of the Giant Panda Nature Reserve and its external buffer zone has changed considerably as the process of social economic development and population exchange continues. Among the major landscape changes is the increase in cultivated land and the decrease in forest land caused changes in planting behavior, changes in the forest coverage rate and the vegetation index caused by fuelwood utilization and logging behavior, and the rise of umbrella species brought about by changes to the giant panda population. The factors driving the dynamic change to landscape patterns in the Giant Panda Nature Reserve and its external buffer zone can be divided into two categories: natural factors and human factors. According to the changes in these areas from 2000 to 2017, this study assessed land cover data in 2015 and 2017. The LR model was used to analyze the driving mechanism of landscape pattern change in the Giant Panda Nature Reserve and its external buffer zone. DEM ($X_1$), Slope ($X_2$), Aspect ($X_3$), Water Body ($X_4$), Giant Panda Activity Trace ($X_5$), Farmer Planting Behavior ($X_6$), Firewood Use Behavior ($X_7$), and Road Use ($X_8$) were selected as eight driving factors. DEM ($X_1$), Slope ($X_2$), Aspect ($X_3$), Water Body ($X_4$) and Road ($X_8$) were obtained from remote sensing imagery interpretation. The criterion of Giant Panda Activity Trace ($X_5$) was the discovery point of excrement that verify that giant pandas used the habitat. The data for Farmers' Planting Behavior ($X_6$) and Firewood Use Behavior ($X_7$) came from the questionnaire survey of farmers in which farmers identified their utilizable cultivated land and firewood cutting sites.

As seen in Table 4, the change of habitat landscape pattern from 2015 to 2017 in the study area correlated with topographic factors (DEM, slope, and aspect), land cover (water body), giant panda activity traces, and farmer resource utilization activities (planting behavior and firewood-use behavior). The mean ROC (receiver operating characteristic) of the four scenarios was 0.871, and ROC test values for each scenario were all over 0.850, indicating that the LR model was accurate and sufficiently credible in analyzing the driving factors of giant panda habitat landscape pattern changes in the WWW Nature Reserve.

The key explanatory variables for the evolution of giant panda habitat landscape patterns were giant panda activity traces and household resource use activities (planting behavior and firewood-use behavior). Different labor-force structure changes have created differentiated development trajectories of farmer resource utilization behaviors, influenced by the development trend of human society. This results in differentiation of the landscape structure of giant panda habitats. Therefore, in this part, the paper reveals the phenomenon of giant panda habitat differentiation caused by differences in household resource-use intensity under the four scenarios (ecological priority, coordinated development, partial constraint, and current situation) generated by changes in household resource-use behavior.

**Table 4.** Logistic regression analysis results for habitat landscape pattern evolution in the WWW Nature Reserve, 2015 to 2017.

| Variable | Ecological Priority | | Coordinated Development | | Partial Constraint | | Current Situation | |
|---|---|---|---|---|---|---|---|---|
| | Regression Coefficient | Standard Deviation | Regression Coefficient | Standard Deviation | Regression Coefficient | Standard Deviation | Regression Coefficient | Standard Deviation |
| $X_1$ | 0.003 | 0.000 | 0.003 | 0.000 | 0.003 | 0.000 | 0.003 | 0.000 |
| $X_2$ | 0.012 | 0.001 | 0.012 | 0.001 | 0.015 | 0.001 | 0.013 | 0.001 |
| $X_3$ | −0.003 | 0.000 | −0.003 | 0.000 | −0.003 | 0.000 | −0.003 | 0.000 |
| $X_4$ | 1.572 | 0.431 | 1.599 | 0.425 | 1.591 | 0.420 | 1.577 | 0.421 |
| $X_5$ | 24.445 | 0.634 | 13.844 | 0.430 | 18.269 | 0.467 | 2.468 | 0.366 |
| $X_6$ | −8.222 | 0.291 | −3.747 | 0.415 | −9.783 | 0.526 | −6.896 | 0.381 |
| $X_7$ | −6.765 | 0.200 | −3.907 | 0.380 | −4.878 | 0.505 | −4.180 | 0.422 |
| $X_8$ | −2.647 | 0.131 | −2.185 | 0.176 | −2.517 | 0.238 | −0.230 | 0.179 |
| Constant | −6.995 | 0.092 | −6.617 | 0.089 | −5.393 | 0.083 | −8.004 | 0.090 |
| ROC | 0.854 | | 0.867 | | 0.875 | | 0.889 | |

The landscape level of the giant panda habitat in 2025 under the ecological priority scenario is generally higher than that under the partial constraint, coordinated development, and current situation scenarios. In terms of planting behavior, the contribution rate of planting (−0.8222) in the ecological priority scenario of fully implementing the conversion of farmland to forest was generally higher than that in other scenarios. In the partially constrained scenario, local governments and protected areas have stronger control over farmers' planting behavior than in the coordinated development scenario. Therefore, the contribution of cultivation behavior under the partial-constraint scenario (−9.783) in terms of habitat improvement for giant pandas was higher than under the coordinated development scenario (−3.747). Based on the feminization and aging of the household labor force, the linear change under the influence of the control system leads to many peasant households withdrawing from planting. This caused cultivation behavior (−6.896) to drive the optimization of panda habitat levels under this scenario.

In terms of fuelwood use behavior, the driving force (−6.765) under the ecological priority scenario, where forest felling was prohibited, was higher than in the other three scenarios (24.445) for improvement in the level of quality of giant panda habitats. The intensity of fuelwood-use behavior was the direct driving factor for habitat improvement of giant pandas under partial constraint. The clean energy that farmers can obtain when the utilization of forest resources is constrained changes the structure of farmers' energy use by replacing fuelwood. In a coordinated development scenario, however, there are still readily available and cheap clean-energy sources to replace fuelwood. However, farmers have a large area of commercial forest from which to obtain firewood and continue traditional energy use (including smoking bacon, use for fire, bamboo shoot drying, etc.). Therefore, the degree of disturbance for fuelwood utilization in giant panda habitats is still high under this scenario. In the current situation, after controlling for institutional factors, farmers tend to prefer non-agricultural business activities, such as eco-tourism; the trend of feminization and aging of the labor force continues to weaken labor force supply for resource utilization. This change indirectly reduces the intensity of farmers' fuelwood utilization and promotes improvement in giant panda habitat quality.

*3.4. Prediction of Landscape Ecological Patterns*

Based on the CA-Markov model, the landscape pattern distribution of the WWW Nature Reserve in 2025 was analyzed and landscape-type prediction results were obtained.

As seen in Table 5, there are differences in habitat landscape pattern changes under the different scenarios. Under the ecological priority scenario, all land use types are agglomerated as forest land. Under the coordinated development and partial-constraint scenarios, any artificial surface (construction land and unused land) was transferred to woodland and grassland. Under the current situation, these elements represent transformation of

part of the arable land, construction land, unused land, and other disturbed land to forest, grassland, water, and other categories.

**Table 5.** Habitat landscape-type change in the Wawushan Nature Reserve under the different scenarios in 2025.

| Land Use Type | | Ecological Priority | Coordinated Develop-ment | Partial Con-straint | Current Situation | Land Use Type | | Ecological Priority | Coordinated Develop-ment | Partial Con-straint | Current Situation |
|---|---|---|---|---|---|---|---|---|---|---|---|
| Farmland | Area (km²) | 0 | 0.2704 | 0.1036 | 1.5159 | Water | Area (km²) | 0.3351 | 0.3699 | 0.3463 | 0.3853 |
| | Proportion (%) | 0 | 0.07 | 0.03 | 0.42 | | Proportion (%) | 0.09 | 0.10 | 0.09 | 0.11 |
| | ROC (%) | −100.00 | −81.94 | −93.08 | 1.22 | | ROC (%) | −9.41 | 0.00 | −6.40 | 4.16 |
| Woodland | Area (km²) | 362.5992 | 361.4209 | 361.7085 | 359.6832 | Construction land | Area (km²) | 0 | 0 | 0 | 0.2921 |
| | Proportion (%) | 99.37 | 99.05 | 99.13 | 98.57 | | Proportion (%) | 0 | 0 | 0 | 0.08 |
| | ROC (%) | 0.71 | 0.38 | 0.46 | −0.10 | | ROC (%) | −100.00 | −100.00 | −100.00 | 116.40 |
| Grassland | Area (km²) | 1.9666 | 2.8398 | 2.7427 | 2.7341 | Unutilized land | Area (km²) | 0 | 0 | 0 | 0.2904 |
| | Proportion (%) | 0.54 | 0.78 | 0.75 | 0.75 | | Proportion (%) | 0 | 0 | 0 | 0.08 |
| | ROC (%) | −30.75 | −0.00 | −3.42 | −3.73 | | ROC (%) | −100.00 | −100.00 | −100.00 | 2400.20 |

Under the ecological priority scenario, forest area increased significantly (change rate: 0.71%) and the regional forest coverage rate increased to 99.37%. With the implementation of the system of returning farmland to forest and forbidding humans to enter the protected area, vegetation types in the region have been developing at a higher level. Grassland is shifting to woodland and the visible water surface (without canopy cover) is shrinking. The area of forested land has risen sharply. The area of disturbed land, such as cultivated land, construction land, and unused land, is reduced to zero under the constraints of the system.

In contrast, the increase in forested area and the forest coverage rate reached 0.46% and 99.13% in the partially constrained scenario with slightly reduced regulation intensity. Artificial surfaces turned to woodland, grassland, water, and other quality habitats for pandas. Under this scenario, the planting of characteristic agricultural products is still allowed. By promoting a new round of returning-farmland-to-forest projects, the area of arable land reduces by 93.08% to adapt to the conservation target of giant panda habitats. With changes to the internal labor force structure and to resource utilization behavior and coupled with restrictions on the external labor force system, an abundance of grassland, water areas, and other land types is transforming the region into woodland. The ratio of grassland and water reached 3.42% and 6.40%, respectively.

In the coordinated development scenario, to reduce the contradiction between protection and development, farmers are still supported to plant characteristic crops and retain collective forests. However, 0.38% of forestland was transformed and 81.94% of cultivated land was converted. The results show that farmers can still support improvement in the regional ecological environment in the context of a moderate alleviation of the contradiction to achieve harmony between development (the cultivation of white tea, yams, and Coptis requires the use of native habitats) and conservation.

In the current scenario, the area of forestland and grassland decreased by 0.10% and 3.73%, respectively, while the area of arable land increased by 1.22%. Neither external institutions nor internal labor structures changed. The habitat level of the giant panda is predicted to decrease significantly.

*3.5. Habitat Quality Simulation*

Figure 5 and Table 6 report the simulated results of habitat quality in the Wawushan Nature Reserve in 2025. The results show that under the ecological priority scenario, the proportion of HH species was 99.37% (an increase of 0.71%, compared with 2017), and

the area of LH and LL species was zero. The overall habitat level of the region under high-intensity control greatly improved. In the coordinated development and partially constrained scenarios, the proportion of living HH species reached more than 99% (99.05% and 99.13%, respectively), increasing by 0.38% and 0.46%, compared with 2017. Among them, the proportion of LH students decreased to 0.07% and 0.03% (81.94% and 93.08%, respectively, compared with 2017), and the area of LL students decreased to zero. These results indicate that appropriately reserving part of the cultivated land for characteristic planting and granting farmers appropriate logging permission to gradually improve the energy utilization structure can drive coordination between regional conservation and development and improve regional giant panda habitat levels. However, in the current situation, the proportion of HH and HL decreased by 0.10% and 3.72%, compared with 2017, while the proportion of low-grade land increased 2.97 times. The results indicate that maintaining the current levels will aggravate disturbance from households' resource utilization behavior of giant panda habitats. With the passage of time, farmers' dependence on resources increases exponentially [19,21–25], which will pose a serious threat to panda habitats and break the balance between conservation and development.

**Table 6.** Habitat quality of the Wawushan Nature Reserve under the different scenarios in 2025.

| Land Use Type | | Ecological Priority | Coordinated Development | Partial Constraint | Current Situation |
|---|---|---|---|---|---|
| HH | Area (km²) | 362.5992 | 361.4209 | 361.7085 | 359.6832 |
| | Proportion (%) | 99.37 | 99.05 | 99.13 | 98.57 |
| HL | Area (km²) | 1.9666 | 2.8398 | 2.7427 | 2.7341 |
| | Proportion (%) | 0.54 | 0.78 | 0.75 | 0.75 |
| M | Area (km²) | 0.3351 | 0.3699 | 0.3463 | 0.3853 |
| | Proportion (%) | 0.09 | 0.10 | 0.09 | 0.11 |
| LH | Area (km²) | 0 | 0.2704 | 0.1036 | 1.5159 |
| | Proportion (%) | 0 | 0.07 | 0.03 | 0.42 |
| LL | Area (km²) | 0 | 0 | 0 | 0.5825 |
| | Proportion (%) | 0 | 0 | 0 | 0.16 |

Under the four scenarios, the areas with low habitat quality were construction land and cultivated land, while the areas with high habitat quality were woodland, shrubland, grassland, and wetland. If internal and external factors can continue to drive labor force feminization and aging and continue to develop after 2025 under the scenario of ecological priority, coordinated development, and partial constraints, the area of giant panda micro-habitat, such as forest land and shrubland, will continue to increase, while the area of construction land and cultivated land will continue to decrease.

### 3.6. Effects of Farmers' Preference for Resource Use on Habitat Evolution of Giant Pandas Habitat

The aforementioned land cover status and simulation results show that construction land, arable land, shrubland, and other land types remain that are closely related to the production and lifestyle of farmers surrounding the Wawushan Nature Reserve. This paper, therefore, takes Changhe Village (forestry resource-dependent community) and Heishan Village (planting-dependent community) of Wawushan Town, Hongya County, Meishan City, Sichuan Province, both of which are closely related to the Wawushan Nature Reserve, as typical communities. By combing the historical development context, the current situation of resource utilization, and farmers' expectation of resource utilization in Changhe Village, the rationality of regional land use/cover change at present and in the future was explained, revealing the evolution direction of regional giant panda habitats.

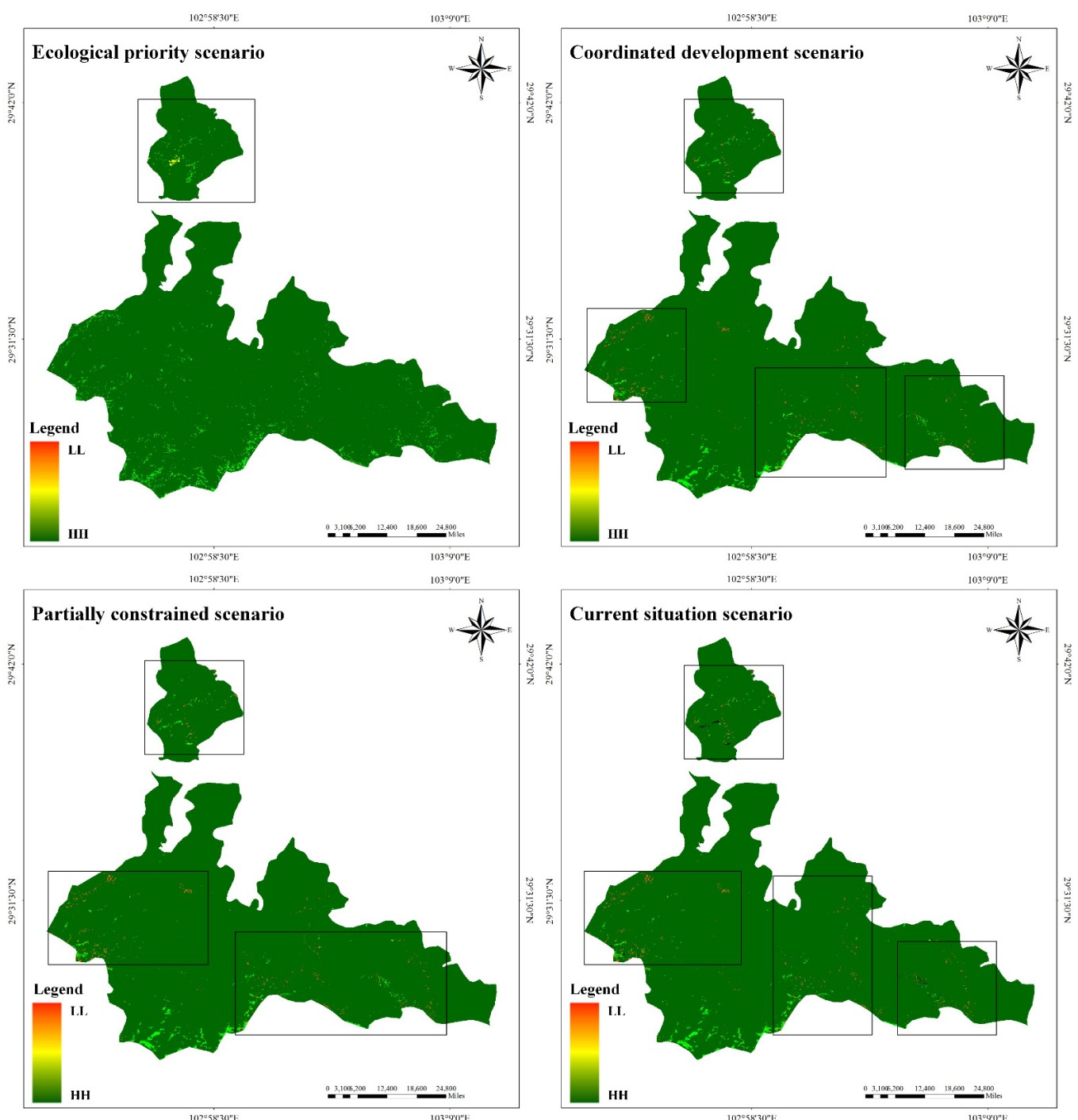

**Figure 5.** Habitat quality of Wawushan Nature Reserve under the different scenarios in 2025. Notes: ① The proportion of HH land under the ecological priority scenario was the highest, while the proportion of HH land under the current situation scenario was the lowest. Under the ecological priority, coordinated development, and partial-constraint scenarios, the proportion of LL land type area was 0, while under the current situation, it was higher. ② Under the four scenarios, the changes of habitat quality occurred mainly in the northern experimental area, in the area adjacent to the Daxiangling Nature Reserve in the west, and at the junction of Leshan city in the south. The difference of habitat quality was greatest in the experimental area. ③ The ranking of habitat quality under the different scenarios in the Wawushan Nature Reserve in 2025 was ecological priority scenario > partially constrained scenario > coordinated development scenario > current situation scenario.

### 3.6.1. Forestry Resource-Dependent Communities: A Case Study of Changhe Village

1.    Community Profile

Changhe Village is surrounded on three sides by the Wawushan Nature Reserve. According to the terrain, an iron gate is set at the entrance of the village as a barrier between the protection area, Changhe village, and the outside world. The special geographical position and topographic structure influence the landscape structure of the Wawushan Nature Reserve. It is a typical rural-household labor force with a feminized and aging community.

2.    Resource Use Conflict: History and Present Situation

In the field of natural resource utilization, the fuelwood utilization behavior of Changhe Village households caused the most serious disturbance to giant panda habitats. The interference process has a historical pattern of "annual decline and short-term rebound". The root cause of this change lies in the establishment and closure of small hydro-power stations in the Wawushan area. The cancelation of the free electricity policy brought about a psychological loss to farmers who increased the use of free fuelwood to reduce electricity costs, resulting in a rebound in fuelwood-use intensity. Farmers are increasingly dependent on fuelwood for cooking and heating, leading to changes in the regional forest cover structure (Figure 6 and Table 7).

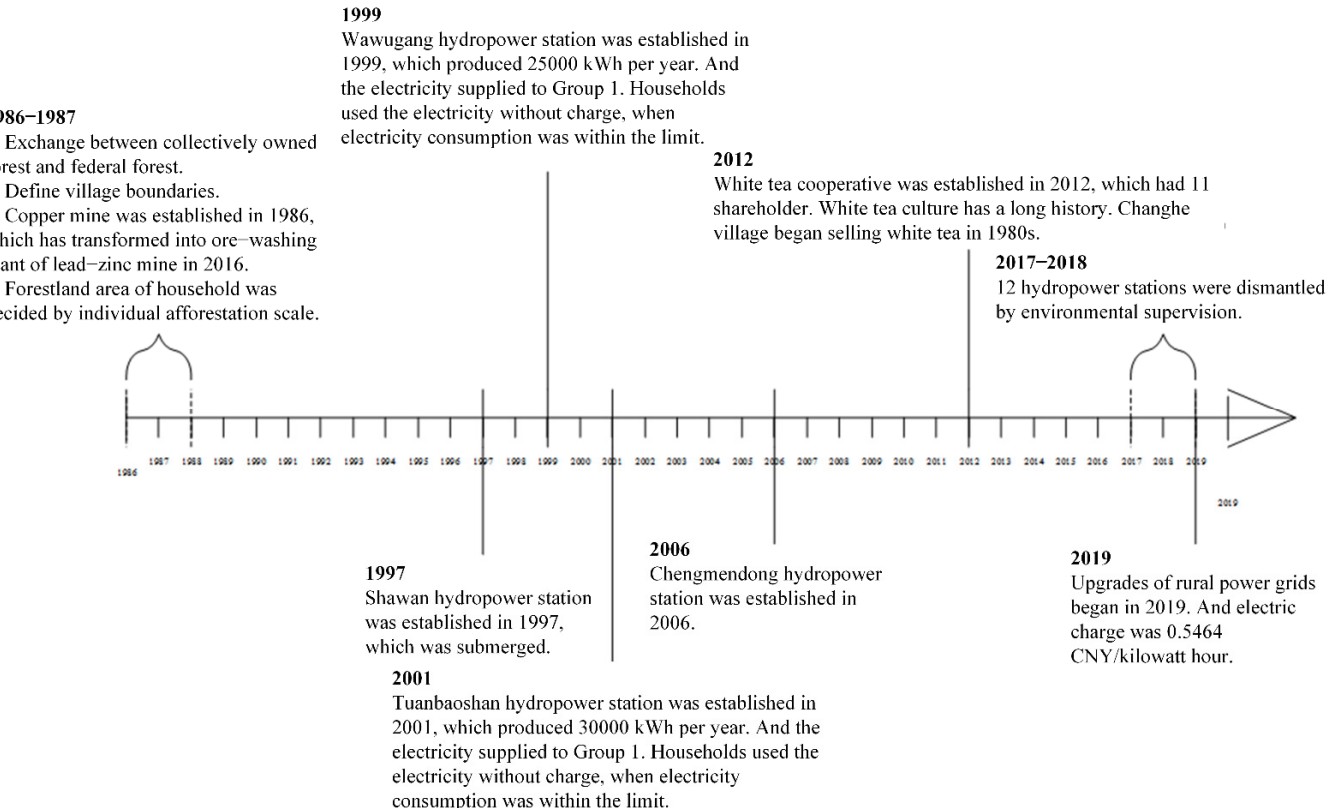

**Figure 6.** History of Changhe Village, Wawushan Town. Data source: description of participants in the experiment.

At present, farmers in Changhe Village are faced with the contradiction of the demand for fuelwood and control of natural resources. In terms of forestland, Changhe Village has returned farmland to forest; there is no arable land in the region. In addition, most forestland in Changhe Village is classified as ecological public welfare forest, and a small amount of commercial forest still lacks any indication of felling. On the one hand, this is not conducive to forest management (cryptomeria fortuna needs thinning to achieve scientific forest management). On the other hand, it aggravates the contradiction between farmers' access to resources for their livelihood (regional farmers use forest management

as their source of livelihood, which is also an important source of firewood) and protection. Therefore, the adverse selection of farmers felling behavior (felling firewood in the hinterland of the giant panda reserve in the east, west, and south of the village) cannot be avoided in the current contradictory situation (as shown in Figure 7). It is impossible to avoid land use/cover change around the reserve.

**Table 7.** Events of Changhe Village, Wawushan Town.

| Year | Event |
|---|---|
| 1986–1987 | 1. Exchange between collectively owned forest and federal forest;<br>2. Defined village boundaries;<br>3. Copper mine was established in 1986 and transformed into an ore-washing plant for lead-zinc mine in 2016;<br>4. Forestland area of households was determined by individual afforestation scale. |
| 1997 | Shawan hydropower station was established in 1997 and subsequently submerged. |
| 1999 | Wawugang hydropower station was established in 1999, which produced 25,000 kWh per year. The electricity was supplied to Group 1 households who used the electricity without charge when electricity consumption was within limits. |
| 2001 | Tuanbaoshan hydropower station was established in 2001, which produced 30,000 kWh per year. The electricity was supplied to Group 1 households who used the electricity without charge when electricity consumption was within limits. |
| 2006 | Chengmendong hydropower station was established in 2006. |
| 2012 | The white tea cooperative was established in 2012 with 11 shareholders. White tea culture has a long history. Changhe Village began selling white tea in the 1980s. |
| 2017–2018 | Twelve hydropower stations were dismantled under environmental supervision. |
| 2019 | Upgrades of rural power grids began in 2019. The electric charge was 0.5464 CNY/kilowatt hour. |

Data source: description of participants in the experiment.

3. Farmers' Coping Strategies and their Resource Utilization under Different Regulatory Intensities

The scenario experiment invited nine villagers (six male and three female) whose average age is over 50 with different social backgrounds to participate in a workshop. They included a reserve manager, three village cadres, a power station representative, two cooperative representatives, and two villagers. According to the nine participants, three factors concerned them: "whether trees can be cut down", "it is beneficial to close the hydro power station", and "whether the public welfare forest can be used". Based on these factors, the combination of the needs to establish a national park for giant pandas affects farmers' general control of resource utilization. According to four of the farmers, this constituted controlling the degree of cognitive structure, namely, having a strong ecological priority (control), coordinating development scenarios (controlling the pine), some constraint with moderate intensity (control), and maintaining the status quo.

The researchers then consulted farmers about countermeasures under the different scenarios and designed strategies for the four scenarios in Changhe Village, according to the clustering principle by sorting farmers' opinions. The strategies focused on farmers' firewood utilization behaviors (as shown in Table 8). Based on the above three factors identified by the participants and the establishment of general control measures for household resource utilization in the Giant Panda National Park, four scenarios are constructed, according to the recognition of households of the degree of control. The scenarios were: ecological priority (highly regulated), coordinated development (least regulated), partially constrained (moderate intensity of regulation), and status quo. The researchers consulted farmers about the countermeasures in different scenarios and constructed strategies for four scenarios in Changhe Village according to the clustering principle by sorting farmers' opinions. Response strategies focus on household fuelwood-use behavior (Table 8).

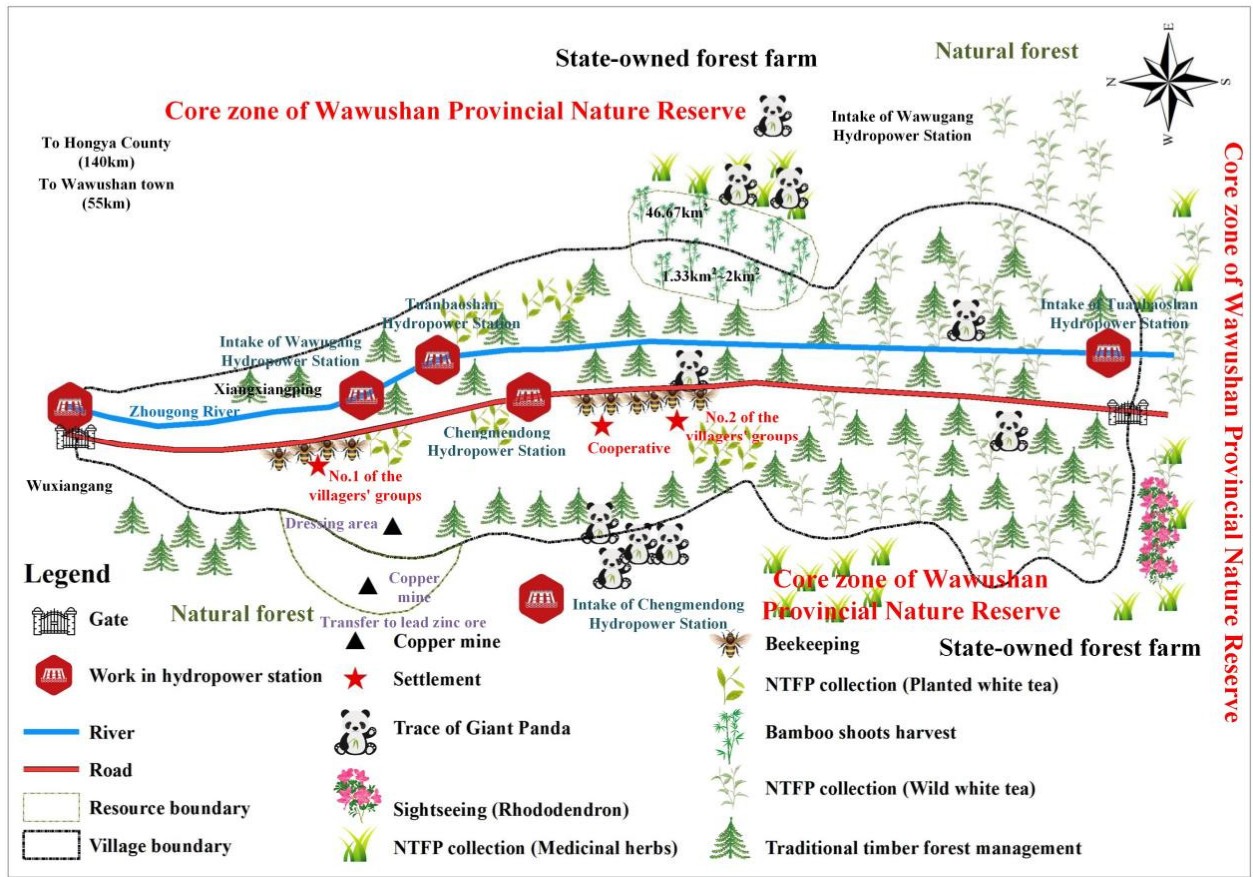

**Figure 7.** Resource utilization status of Changhe Village, Wawushan Town (Hand-drawn Map). Note: The firewood-logging activities of farmers in Changhe Village extend to the eastern, western, and southern protected areas. As there are many giant panda movement tracks in Changhe Village and its surrounding areas, farmers' firewood-cutting behavior in this area will significantly disturb giant panda habitats. Data source: descriptions of participants in the workshop experiment.

**Table 8.** Coping strategy for farmers' resource-utilization behavior under the different scenarios.

| Category | Scenario Content |
|---|---|
| Scenario 1: Ecological priority | |
| Content | • One-third of forestland is included in the core area of the national park (part of the public welfare forest is included). Human access is forbidden to this part of the forest; human use behavior is prohibited. <br> • In the ecological public welfare forest, felling is banned in the current stage. Harvesting can only be renewed after 35 years, and the harvesting area is limited. <br> • Commercial forests do not allow commercial logging and contracts can only be renewed. <br> • Firewood cutting, planting, and other resource-utilization activities that interfere with the habitat of giant pandas are prohibited. |
| Coping strategies | • Through whole-village migration to obtain more available livelihood resources (food, energy) and to achieve convenient employment, school access, and medical treatment. The migration of the whole village should be realized through resource purchase with compensation via capital and follow-up life-skills training. <br> • In the utilization of forest resources, selective logging can be adopted to reduce the proportion of forestry operations in livelihood assets. |

**Table 8.** *Cont.*

| Category | Scenario Content |
|---|---|
| **Scenario 2: Coordinated development** | |
| Content | • Not all forestlands are included in the core area of the national park; the ecological public welfare forest allows selective cutting for species care-raising, and the target of commercial forest cutting meets the demand.<br>• In terms of planting, the scale of white tea planting can be expanded.<br>• In terms of natural resource utilization, villagers only are allowed to plant bamboo shoots and collect herbs. This can not only protect the rights and interests of farmers in the village to use resources but can also prohibit the commercial development of resources and the intensity of interference with giant panda habitats.<br>• In terms of eco-tourism, farmers will be allowed to transform their agritainment. Clear water and green mountains become gold and silver mountains.<br>• Under the principle of volunteerism, the protection zone reduces the requirement to give ecological employment positions to villagers.<br>• According to villagers' wishes, 33 hydropower stations were closed. Farmers are allowed to raise ecological fish and carry out splashing and drifting projects. |
| Coping strategies | • Farmers can increase the proportion of electricity and food they buy as their livelihood capital increases. Reduces the use of fuelwood and farmland.<br>• Farmers can obtain firewood through selective cutting of forestland to avoid entering the reserve to cut firewood. |
| **Scenario 3: Partial constraint** | |
| Content | • Only a small amount of forest land is allowed to be designated as the core area of the national park where human access and use is forbidden<br>• Ecological public welfare forest access is banned and the commercial forest cutting index is far lower than demand.<br>• The scale of white tea planting is limited.<br>• Farmers are allowed to collect medicine and plant bamboo shoots in the general control area for natural resource utilization, which is not exclusive.<br>• New agritainment of ecological tourism is prohibited to reduce the ecological burden caused by the human footprint.<br>• Protected areas can provide some ecological jobs for farmers, but there are requirements on age, gender, physical capacity, and other aspects.<br>• Hydropower stations are closed and blank-period watershed development in the system is temporarily shelved. |
| Coping strategies | • The compensation for commercial forests for forestland resource utilization is increased by replacing bamboo forests, and compensation for the ecological public welfare forest was increased.<br>• Close the hydropower station to allow breeding of ecological fish, rafting, and other projects; otherwise, do not close the hydropower station to benefit from the electricity discount.<br>• Hope to allow ecotourism development. |
| **Scenario 4: Current situation** | |
| Content | • The ecological public welfare forest cannot be felled within 35 years; the felling cycle of the commercial forest has not been reached; and the quality of farmers' commercial forest is different.<br>• In the planting industry, the quality of white tea picking is different, lacking industry standards.<br>• Planting of bamboo shoots and harvesting herbs as natural resources are allowed (sleeping on the mountain is prohibited). It is forbidden to cut firewood in ecological public welfare forests and protected areas.<br>• There are no farmhouses in Changhe Village.<br>• There are only temporary forest rangers in Changhe Village.<br>• There are three small hydropower stations in operation in the village. |
| Coping strategies | • The reason that farmers can choose to cut commercial forests is that it is helpful for ecological restoration and to meet the needs of farmers for firewood utilization. |

Data source: description of participants in the experiment.

In the ecological priority scenario, the use of cultivated land and woodland resources are prohibited. Farmers hope to improve their living environment by moving the whole village and allowing selective felling for forest tending. On the one hand, this strategy is motivated by directly cutting off traditional sources of livelihood. On the other hand, consideration is given to the sharp decline of available labor brought about by labor-force feminization and aging (see Table 8).

In coordinated development scenarios, both farmer's resource utilization requirements (the scale of white tea planting was allowed to expand; the collective forest category is not classified as national park; the ecological public welfare forest allows tending of selective, commodity-cutting index forest species to meet requirements) and controls of natural resources (allowing farmers to intensify the use of natural resources) are combined. Thus, the livelihood capital of farmers will increase, and the strategy of increasing the proportion of electricity consumption and food purchase and of reducing firewood and farmland use behavior will be implemented. The reasons for this strategy are as follows: firstly, the structure of the regional labor force is optimized and the male and young-labor force move to non-agricultural activities. Women and the elderly carry out agricultural resource utilization activities within their capacity. At this point, development and protection can be balanced. Secondly, improvement in agricultural and non-agricultural income has enhanced the livelihoods of farmers. Farmers have extra capital to spend on food and clean energy, which optimizes the allocation of labor, provides more leisure, and improves well-being (see Table 8).

The intensity of natural resource regulation in the partial constraint scenario is slightly higher than that in the coordinated development scenario. In this situation, the total amount of natural resources available to farmers is reduced (a small number of collective forests are included in national parks, and the scale of white tea cultivation is limited), the scope of resource utilization activities is restricted (woodlands designated as national parks are off limits to humans), and the direction of livelihood transformation in the region is relatively scarce (new farm entertainment and ecological posts are prohibited and require a high-quality labor force). Farmers hope to improve their livelihood by obtaining ecological compensation and allowing ecological product management. One possible reason for the development of this strategy is that farmers' planting and fuelwood-utilization behaviors are subject to stricter institutional constraints, and thus farmers' resource utilization is limited. Farmers seek to operate ecological products (eco-fish farming, eco-tourism) within the framework of the existing Giant Panda National Park system (guidance on establishing a system of protected natural areas with national parks as the main body) to reduce the impact of institutional pressure and the feminization and aging of the labor force on farmers' livelihood (see Table 8).

In the current situation, it is still in the transitional period between the nature reserve system and the national park system, and the new system has not been enacted. To avoid repeated construction caused by system shock, the local government and the reserve administration in the region adhere to the policy of maintaining the existing system, that is, neither increasing nor reducing farmers' resource utilization activities. The trend of the labor force feminizing and aging fluctuates slightly with the change in the external environment. However, it is difficult to change farmers' dependence on planting and fuelwood. Therefore, the disturbance caused by the resource utilization behavior of farmers to the panda habitat will continue (as shown in Table 8).

4. Habitat Evolution of Giant Pandas under Different Control Intensification

Changhe Village is a typical community with a feminizing and aging household labor force dependent on forestry resources. Based on the above four scenarios, the evolutionary direction of the local regional habitat of the Giant Panda Nature Reserve affected by Changhe Village can be divided into the following four categories:

- Scenario 1: Ecological priority. Under the ecological priority scenario, migration of the whole village can stop the impact of farmers' resource utilization behavior on giant

panda habitats. After a certain period of natural restoration, the land disturbed by humans can be naturally returned to woodland, shrubby, and other high-quality land. In this way, the local habitat quality can be significantly improved, and the simulation results of ecological priority scenarios in the previous paper can be verified.

- Scenario 2: Coordinated development. In the coordinated development scenario, farmers support electricity and food consumption with income from ecotourism operations. In this way, fuelwood and arable land use are reduced, and the contradiction between protection and development is small. The change intensity of resource use behavior is weaker than that of the partial constraint scenario. Consequently, the forest cover in the affected areas will be restored and vegetation indices will increase. The simulation results of the coordinated development scenario above are verified.

- Scenario 3: Partial constraint. In the partial constraint scenario, the resource utilization behavior of farmers is strongly constrained by the system. Farmers are willing to pay for free electricity in exchange for the development of ecological products that women and older workers can accomplish. Improvements in livelihoods are expected to compensate for the shortfall in food and energy consumption. In this way, the negative impact of institutional and labor structure changes on the livelihood of farmers is lessened, and the disturbance to the habitat of giant pandas is reduced. Therefore, the habitat landscape pattern of the affected Giant Panda Nature Reserve should be smoothly transformed into high quality. The simulation results of some constraint scenarios mentioned above are verified.

- Scenario 4: Current situation. In the present situation, farmers hope to selectively cut commercial forests. However, this conflicts with the existing system, meaning that the adverse selection behavior of farmers cannot be avoided. Under the background of feminizing and aging of the peasant labor force, the habitat quality of the regionally influenced Giant Panda Nature Reserve will change significantly in the next few years. The lack of institutional coordination will result in an overall decline in habitat levels (verified in Figure 5), which confirms changes in the habitat quality of giant pandas in the future, under the current situation.

3.6.2. Planting-Dependent Communities: A Case Study of Heishan Village

1. Community Profile

Heishan Village is surrounded on three sides by the Wawushan Nature Reserve. The young and middle-aged workers in the village either go out to work or work in small hydropower stations. The agricultural and forestry production activities in the village are dominated by women and elderly workers. The special topographic structure has a typical effect on the habitat quality of the Wawushan Nature Reserve.

2. Resource Use Conflict: History and Present Situation

In natural resource utilization, farmers' planting behavior in Heishan Village has an impact on giant panda habitats, which is represented by the extension of planting on land in the interior of the reserve and is rooted in habitat selection for Yalian and Coptidis. With the extension of human activities to the hinterland of the reserve, the disturbance to giant panda habitats is intensified by planting activities, which leads to changes in the regional forest cover structure (Figure 8 and Table 9).

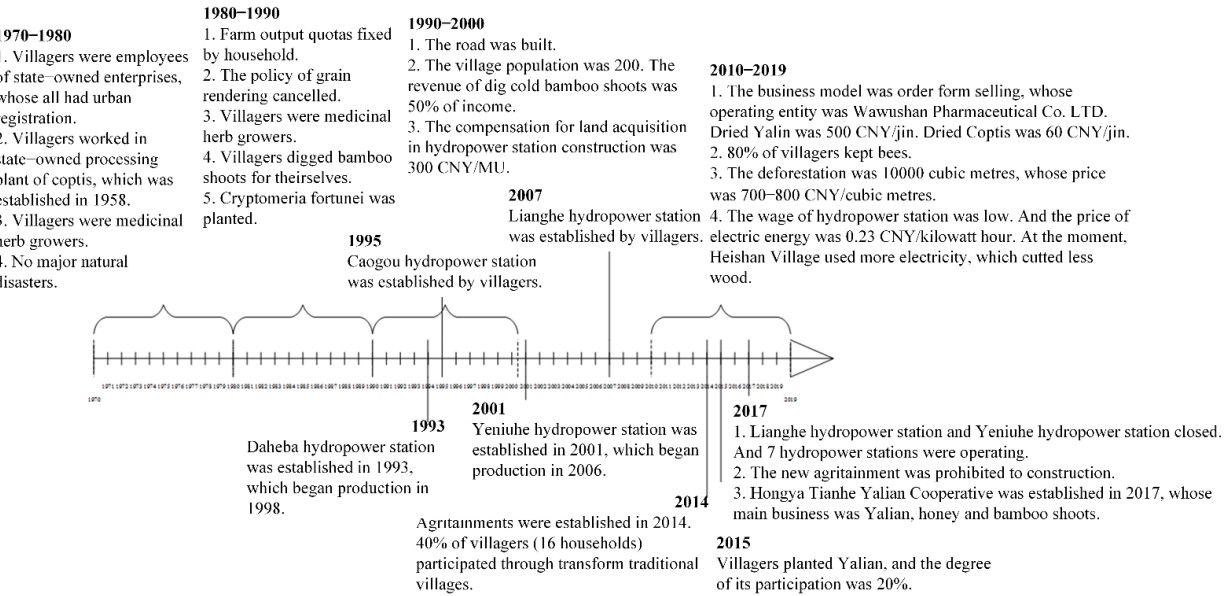

**Figure 8.** History of Heishan Village, Gaomiao Town. Data source: description of participants in the experiment. 2015. Villagers planted Yalian, and the degree of participation was 20%.

**Table 9.** Events of Heishan Village, Wawushan town.

| Year | Event |
|------|-------|
| 1970–1980 | 1. Villagers were employees of state-owned enterprises, who all had urban registration.<br>2. Villagers worked in the state-owned processing plant of Coptis, which was established in 1958.<br>3. Villagers were medicinal herb growers.<br>4. No major natural disasters. |
| 1980–1990 | 1. Farm output quotas fixed by household.<br>2. The policy of grain rendering was cancelled.<br>3. Villagers were medicinal herb growers.<br>4. Villagers dug bamboo shoots for themselves.<br>5. Cryptomeria fortunei was planted. |
| 1990–2000 | 1. The road was built.<br>2. The village population was 200. The revenue for digging bamboo shoots was 50% of income.<br>3. The compensation for land acquisition in hydropower station construction was 300 CYN/MU. |
| 1993 | Daheba hydropower station was established in 1993, which began production in 1998. |
| 1995 | Caogou hydropower station was established by villagers. |
| 2001 | Yeniuhe hydropower station was established in 2001, which began production in 2006. |
| 2007 | Lianghe hydropower station was established by villagers. |
| 2010–2019 | 1. The business model was order form selling, whose operating entity was Wawushan Pharmaceutical Co., Ltd. Dried Yalin was 500 CYN/jin. Dried Coptis was 60 CYN/jin.<br>2. 80% of villagers kept bees.<br>3. The deforestation was 10,000 cubic meters, whose price was 700–800 CYN/cubic meter.<br>4. The wage of hydropower station workers was low. The price of electric energy was 0.23 CYN/kilowatt hour. At the moment, Heishan Village uses more electricity and cuts less wood. |
| 2014 | Agritainments were established in 2014. A total of 40% of villagers (16 households) participated through transforming traditional villages. |
| 2015 | Villagers planted Yalian, and participation was 20%. |
| 2017 | 1. Lianghe hydropower station and Yeniuhe hydropower station closed. Then, hydropower stations were operating.<br>2. Construction was prohibited in new agritainment.<br>3. Hongya Tianhe Yalian Cooperative was established in 2017, whose main business was Yalian, honey, and bamboo shoots. |

Data source: description of participants in the experiment.

Currently, Heishan Village farmers are faced with overlapping habitats for Yalian and Coptis and the habitats of giant pandas. In 2017, there was a new climax of industrial development for Yalian and Coptis in Montenegro Village. Supported by planting technology, product packaging and publicity, and marketing channel construction, the planting activities of Yilian and Rhizoma chinensis expanded from 1500 m to 1900 m above sea level, and the planting range closely surrounded the boundary of Wawushan Nature Reserve (as shown in Figure 9). The giant panda habitat is easily disturbed by planting behavior. In addition, the cultivation of Yalian and Coptis chinensis will deposit toxins in the soil, which will lead to a change in the regional landscape structure and vegetation index, resulting in the deterioration of habitat quality. The land use/cover change of the reserve and its surrounding areas cannot be avoided.

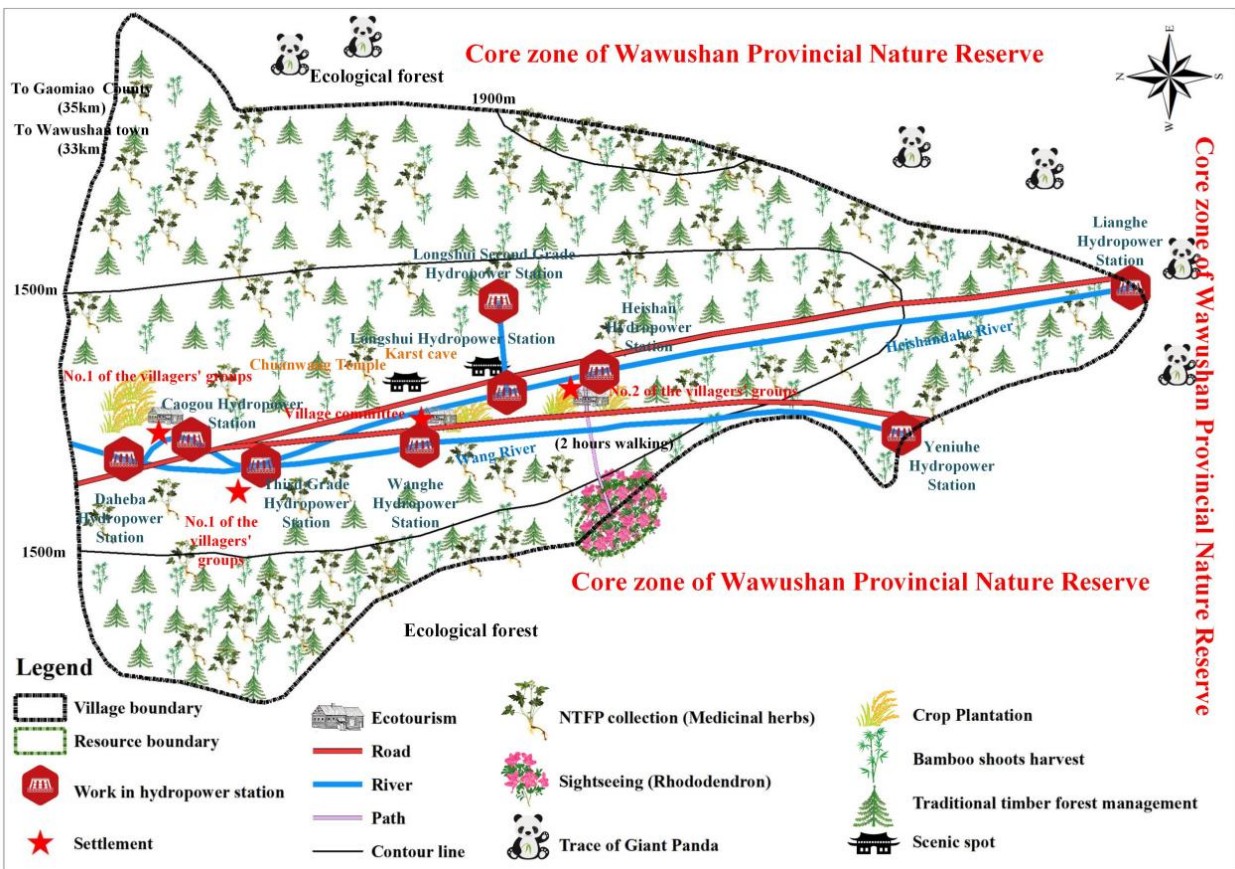

**Figure 9.** Resource utilization status of Heishan Village, Gaomiao Town (Hand-drawn Map). Note: There are traces of farmers' firewood logging activities in Heishan Village extending to the eastern and western protected areas. Since there are giant panda activity tracks around Heishan Village, the planting behavior of farmers in the region will cause considerable disturbance to giant panda habitats. Data source: description of participants in the experiment.

3.     The Coping Strategies of Farmers' Resource Utilization under Different Regulatory Intensity

The scenario experiment invited 13 villagers (12 male and 1 female), with different social roles, to participate. They included a reserve manager, seven village cadres, two forest rangers, and three villagers with an average age of over 50. According to 13 participants, they are most concerned about the planting problem: "whether they can continue to plant Yalian", "whether they can continue to plant Coptis", "whether to move", and three other influencing factors. Four scenarios are constructed based on the above three factors and the general control measures on the utilization of household resources in the Giant Panda National Park being established, as well as acceptance by households of the degree of control. The researchers consulted farmers for countermeasures in different scenarios and

constructed strategies for four scenarios in Heishan Village, according to the clustering principle by sorting farmers' opinions, which focused on farmers' planting behaviors.

In the ecological priority scenario, bamboo-shoot planting, medicinal-drug planting, and wood harvesting were prohibited in the region to reduce the impact of human resource utilization on giant panda habitats. Farmers in Heishan Village wanted to seek land-loss security to maintain their livelihoods. The motivation for this strategy was caused by institutional constraints and changes in the labor structure (see Table 10).

**Table 10.** Coping strategies for farmers' resource-utilization behavior under the different scenarios.

| Category | Scenario Content |
|---|---|
| Scenario 1: Ecological priority | |
| Content | • Planting bamboo shoots, planting medicines, and cutting wood are forbidden in the region to reduce the disturbance of human resource utilization activities on giant panda habitats. |
| Coping strategies | • We will provide basic living allowances for rural households and compensation and insurance to rural households that have lost their sources of livelihood (for land-lost farmers, individuals will pay 40% of the endowment insurance and medical insurance, while the government will pay 60%).<br>• Provide ecological jobs for farmers who have lost their livelihoods. |
| Scenario 2: Coordinated development | |
| Content | • Technological breakthroughs shorten the recovery time of Yalian and Coptis and vigorously develop Heishan flagship ecological products (Yalian series products).<br>• Villagers are allowed to reasonably plant bamboo shoots and medicines on their own forest land for natural resource utilization. This not only protects the rights and interests of farmers in the village to use resources, but also prohibits excessive commercial resource development that intensifies interference to the giant panda habitats. |
| Coping strategies | • Reasonable planning of Yalian and Coptis production under the framework of ecological protection to minimize the impact on panda habitats.<br>• Build Yalian scientific research cooperation base to extend the industrial chain.<br>• Protect the Yalian base, strictly prohibit outside appropriation of water and soil resources. |
| Scenario 3: Partial constraint | |
| Content | • The technological breakthrough made the time for the resumption of cultivation of Yalian and Coptis about five years and encouraged the development of ecological products (Yalian series products).<br>• Villagers are allowed to reasonably plant bamboo shoots and medicines to avoid the over-commercialized resource utilization mode to reduce interference on giant panda habitats. |
| Coping strategies | • To develop the collective economy of the village, carry out joint management, implement community co-management, and ensure orderly development to avoid excessive commercial resource utilization mode for natural resource utilization to reduce interference to the habitat of giant pandas. |
| Scenario 4: Current situation | |
| Content | • The technical bottleneck is still not resolved for the Yalian and Coptis reseeding cycle of 30 to 50 years. Ecological products (Yalian lotus tea, Yalian toothpaste, Yalian mask) have started to operate.<br>• Villagers on their own forest land rationally plant bamboo shoots and Chinese herbal medicines for natural resource utilization. |
| Coping strategies | • Maintain the existing Yalian and Coptis planting scale commits to the development of Yalian series products. The protection of Chinese herbal medicine habitats and giant panda habitats was considered. |

Data source: description of participants in the experiment.

In the coordinated development scenario, the region tries to balance protection and development in the rational use of natural resources to achieve a win–win situation between household income growth and habitat improvement for giant pandas. The main reasons

for this phenomenon are as follows: firstly, similar to Changhe Village, the increase in household income can drive farmers to give up fuelwood use and increase the consumption of alternative energy; secondly, ecological planting needs to create native habitats. To obtain higher incomes from the cultivation of Yalian and Coptis, farmers must explore strategies to improve the water and soil environments and create ecological planting environments to reduce disturbance to the habitats of giant pandas while improving the cultivation environment; thirdly, the weakening of the labor force is an important motivation for coordinating development and protection (see Table 10).

In some constrained scenarios, the intensity of resource utilization control can stimulate farmers' exploration of Yalian planting technology (improving arable land productivity). It effectively alleviates the impact of the contradiction between protection and development of farmers' livelihoods. This includes the development of the village collective economy, the extension of joint management, implementation of community co-management, and orderly development. Thus, a resource utilization model can be developed to avoid excessive commercialization and reduce the interference of planting behavior on the habitat of giant pandas. The reason for the development of this strategy is that the change of system and labor structure can reverse the change of farmers' resource utilization behavior and further advance ecological management (as shown in Table 10).

In the current situation, the transitional period between the nature reserve system and the national park system is still taking place. The status quo has been maintained in all areas of the region with no new or reduced projects. However, maintaining this state for a long time will magnify the impact of non-institutional factors on giant panda habitats and fail to effectively avoid farmers' dependence on natural resources, which may lead to a decline in the level of giant panda habitats (as shown in Table 10).

4. Habitat Evolution of Giant Pandas under Different Control Intensities

Heishan Village is a typical community with a feminizing and aging labor force with planting-industry dependence. Based on the above four scenarios, the evolution of local-regional habitats in the Giant Panda Nature Reserve affected by this community can be divided into the following four categories:

- Scenario 1: Ecological priority. Under the strictest conditions, a ban on the cultivation behavior of Yalian and Coptis could stop the impact of farmers' cultivation behavior on giant panda habitats. This results in a certain period of natural restoration of habitats in the boundary areas of the reserve, which verifies the simulation results of the ecological priority scenarios described above.

- Scenario 2: Coordinated development. The scientific and reasonable planning of Yalian and Coptis production can reduce the disturbance of planting behavior on giant panda habitats. Thus, it can drive the natural recovery of damaged land types and evolve from low-quality land types with a low vegetation index to high-quality land types; that is, the simulation results of the coordinated development scenarios in the previous section are verified.

- Scenario 3: Partial constraint. In the partial constraint scenario, collective management is conducive to avoiding the adverse selection behavior of individual operators and facilitating the supervision of local governments and conservation departments, thus effectively restricting the impact of planting behavior on giant panda habitats. Therefore, the simulation results of the partial constraint scenario in the previous section are verified.

- Scenario 4: Current situation. Under the current situation, farmers hope to maintain the existing planting scale of Yalian and Coptis and devote themselves to the development of Yalian series products. However, it is impossible to avoid the adverse selection of individual farmers to expand the planting scale. Under the background of a feminizing and aging labor force, the habitat quality of the regionally influenced Giant Panda Nature Reserve will change greatly in the next few years. The lack of institutional coordination will result in an overall decline in the habitat level (verified in Figure 5),

which demonstrates changes in the habitat quality of giant pandas in the future under the current situation.

### 3.6.3. Comparison and Validation of Giant Panda Habitat Evolution

According to the participatory multi-scenario experimental analysis, Changhe Village of Wawushan Town and Heishan Village of Gaomiao Town are typical communities with a feminized and aging labor force. Both farmers' planting behavior and fuelwood-use behavior under different regulatory intensities could contribute to long-term habitat quality improvement with farmers' initiatives and policy support (i.e., the simulation results under ecological priority, coordinated development, and partial constraint scenarios). In contrast, under the status quo scenario and due to the adverse selection decisions of farmers, when the labor force is non-feminine and non-aging, the intensity of planting behavior and fuelwood utilization will be intensified, leading to deterioration of the habitat quality of giant pandas in the future (the simulation results of the status quo scenario above).

Through sorting the above-scenario experiments, the results in Table 11 were obtained. The results show that with the background of a feminizing and aging household labor force, the simulation results in Figure 5 can be verified in the context of ecological priority, coordinated development, partial constraints, and the current situation. To sum up, the simulation results mentioned above have been verified here, indicating that the simulation effect is in line with the actual situation in the research region, can be recognized by regional farmers, and passes the test.

**Table 11.** Comparison and verification of direction of evolution of giant panda habitats.

| Forestry Resource-Dependent Communities | | Planting-Dependent Communities | |
|---|---|---|---|
| **Category** | **Data Validation** | **Category** | **Data Validation** |
| **Scenario 1: Ecological priority** Firewood logging is prohibited. | Figure 5 Ecological priority | **Scenario 1: Ecological priority** No cultivation at all. | Figure 5 Ecological priority |
| **Scenario 2: Coordinated development** Commodity forest felling index meets demand and firewood comes from selective felling of commodity forest. | Figure 5 Coordinated development | **Scenario 2: Coordinated development** Reasonable planning of Yalian and Coptis production to minimize the impact on panda habitats and extend the industrial chain. | Figure 5 Coordinated development |
| **Scenario 3: Partial constraint** To give up free energy in exchange for ecotourism development, reduce the use of firewood. | Figure 5 Partial constraint | **Scenario 3: Partial constraint** To reduce disturbance to the habitats of giant pandas, joint operation, and orderly development should be carried out to avoid an excessive commercial resource utilization mode. | Figure 5 Partial constraint |
| **Scenario 4: Current situation** The contradiction between protection and development still exists and fuelwood harvesting cannot be avoided. | Figure 5 Current situation | **Scenario 4: Current situation** Maintain the existing cultivation scale of Yalian and Coptis, but individual farmers cannot avoid adverse selection. | Figure 5 Current situation |

Data source: Collated according to experimental results.

## 4. Conclusions and Discussion

From 2015 to 2017, key nodes in the study area changed the resource utilization intensity of farmers, resulting in changes to the landscape pattern of giant panda habitats in the region. Therefore, based on land cover data for 2015 and 2017, and by combining natural and social factors, this research conducted a habitat simulation based on the CA-Markov model and the logistic regression model. The simulation results showed that the habitat quality of Wawushan Nature Reserve under the ecological priority scenario was relatively high in 2025, and the area of high quality significantly increased, compared with 2015 and 2017. The area of high habitat quality also increased under the coordinated development and partially constrained scenarios. In the current scenario, the area of low habitat quality increased significantly, while the overall habitat quality decreased, compared to 2017. At the same time, in the four scenarios, the areas with low habitat quality were mostly construction and cultivated land with high degrees of human disturbance, while the areas with high

habitat quality were woodland, shrubland, grassland, and wetland. These results indicate that with the backdrop of a feminizing and aging labor force, the positive response of households and institutions can jointly contribute to improvement in giant panda habitats. In contrast, the adverse selection and institutional defects of farmers will be amplified, thus, endangering the quality and safety of giant panda habitats.

To further verify the credibility of the research results, the researchers conducted a participatory scenario experiment in Wawushan Nature Reserve in July 2019 to verify agreement between the direction of future panda habitat evolution caused by, firstly, different control-intensity scenarios from the perspective of farmers and, secondly, the habitat change map through quantitative simulation undertaken in this paper. Empirical evidence shows that with the background of a feminizing and aging household labor force, the simulation results can be verified under the conditions of ecological priority, coordinated development, and partial constraints. However, in the current situation, farmers are prone to magnify adverse selection, so that the simulation results under the same constraints mentioned above can be verified. These results indicate that with a feminizing and aging labor force, the positive response of farmers and institutions can jointly promote the habitat improvement of giant pandas. In contrast, the adverse selection and institutional defects of farmers will be amplified, thus, endangering the quality and safety of giant panda habitats.

**Author Contributions:** Conceptualization, Z.S., Y.L. and L.G.; methodology, Z.S.; software, Z.S.; formal analysis, Z.S. and Y.L.; investigation, Z.S., B.W. and Y.L.; data curation, B.W.; writing—original draft preparation, Z.S., B.W. and Y.L.; writing—review and editing, Z.S., B.W. and W.X.; visualization, Z.S.; supervision, L.G.; project administration, L.G.; funding acquisition, Y.L. and L.G. All authors have read and agreed to the published version of the manuscript.

**Funding:** This research was funded by Ministry of Education, Humanities and Social Sciences Project (Grant number: 22YJCZH150) and National Nature Science Foundation of China (Grant numbers: 71761147003).

**Institutional Review Board Statement:** Not applicable.

**Informed Consent Statement:** Informed consent was obtained from all subjects involved in the study.

**Data Availability Statement:** Not applicable.

**Acknowledgments:** We are grateful to Forestry and Grassland Administration of Sichuan Province, Administration of Wawushan Giant Panda Nature Reserve, Changhe Village and Heishan Village for their invaluable assistance.

**Conflicts of Interest:** The authors declare no conflict of interest.

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
