# Peer review of "Quantifying the Evolution of Giant Panda Habitats in Sichuan Province under Different Scenarios"

_diversity, doi:10.3390/d14100865_

Round 1

Reviewer 1 Report

The article is very interesting and provides significant information for the conservation of the pandas' habitat, through an interdisciplinary analysis. However, it contains information that has not yet been reviewed or designed to be published as an article. The information included in results is extensive, it may be enough to publish several articles or a book. It is recommended to select the most important for this topic, summarize it and include the rest as supplementary material. In the same way, there are many figures and tables.

The document lacks citations in the methodology section (all sections 2.1 and 2.2), it is very important to cite all sources, for example: images, authors of methodologies such as "Land Cover Classification System of Environmental Remote Sensing Monitoring in the Decade of Ecology”, atlas, “natural factors driving habitat change of giant pandas”, “dominant characteristics in the study area”, ASTER GDEM002, fourth giant panda survey report of Sichuan Province, foregoing remote sensing interpretation, road data (National Geographic Information Center), etc.

Describe categories: into 8 first-level categories and second-level categories.

The writing speaks of a questionnaire and field data from which information on "farmers' planting behavior and fuel wood use behavior" was obtained, however, these field data, questionnaires, number of questionnaires applied, etc. are not described.

Why the authors are use only two-year’s difference in the analysis? Justify. It is very little time to demonstrate a change and project it into the future with Markov chains.

Why the authors are you using data from July 2018 to May 2019 to validate the 2015 and 2017 supervised classification? Temporarily, it is not valid, and less so if such a short time interval is being considered, in this case, the minimum changes in land use can be very important.

In this paragraph:

“The framework of this part is "Analysis of Spatio-Temporal Pattern Change of Landscape Pattern - Analysis of Habitat Quality - Analysis of Driving Forces of Landscape Pattern - Simulation of Habitat Quality - Verification of Simulation Results". CA-Markov Model, Logistic Regression Model and Participatory Scenario Method are used in the above five parts respectively.”

The authors refer to five analyzes and three methods and comment that they are applied respectively, but it is not clear which method was used for which analysis (5 and 3 do not correspond). Correct wording.

The authors do not described The Participatory Scenario Method used. Describe it.

The layout of Table 1 is not clear at the bottom left. The codes could be included in Notes to the table.

Figures 2 and 4 do not include the polygon legend of the Wawushan Provincial Nature Reserve Area.

In fig 2, to be able to visualize the changes in both times, it is recommended, to add the same zoom views for both.

The following paragraph and table 2, are found in the results and are part of the method:

“Habitat quality level can reflect the landscape biodiversity and the survival suitability of flagship species. Natural breaks in ArcGIS were used to grade the output habitat quality values ​​into five levels (Table 2) to analyze spatio-temporal differences in habitat quality.”

Figure 4 is very long, select the most important information.

The following paragraph is founded in the results and is part of the method:

“This study was based on land cover data in 2015 and 2017. Logistic regression model was used to analyze the driving mechanism of landscape pattern change in Giant Panda Nature Reserve and its external buffer zone. DEM (X1), Slope (X2), Aspect (X3), Water Body (X4), Giant Panda Activity Trace (X5), Farmers' Planting Behavior (X6), Firewood Use Behavior (X7), and Road (X8) were selected as eight driving factors. DEM (X1), Slope (X2), Aspect (X3), Water Body (X4) and Road (X8) were obtained from remote sensing imagery interpretation.”

Author Response

We are grateful to the editor and reviewers for their positive and constructive comments and criticisms concerning our manuscript, Quantifying the Evolution of Giant Panda Habitats in Sichuan Province under Different Scenarios(diversity-1912699). These comments and criticisms are very helpful for revising and improving our paper. We have made necessary corrections and changes in response to them. Below, our concrete response (indented) follows each specific comment (italicized) of the reviewers. We hope that you will find our revised manuscript is acceptable for publication. Of course, we will make additionally changes if there remain unaddressed or inadequately addressed comments.

Reviewer #1:

  1. The document lacks citations in the methodology section (all sections 2.1 and 2.2), it is very important to cite all sources.

Thanks for your observation and suggestion. In lines 147, 160, 178-186, the citations in the methodology section have been supplemented in the manuscript.

  1. The writing speaks of a questionnaire and field data from which information on "farmers' planting behavior and fuel wood use behavior" was obtained, however, these field data, questionnaires, number of questionnaires applied, etc. are not described.

We appreciate these valuable comments. The data of planting and wood cutting came from field investigation with 64 questionnaires, where lived no more than 200 households according to the authority of WWW Nature Reserve. During the period of field investigation, we invited household to identify the location of planting & wood cutting and the scale of resource utilization.

  1. Why the authors are use only two-year’s difference in the analysis? Justify. It is very little time to demonstrate a change and project it into the future with Markov chains.

Thank you for pointing this out. 2017 is a year of symbolic significance in the study area. In this year, the Ministry of Ecology and Environment inspected on ecological and environmental protection in WWW Nature Reserve. According to the requirements of inspector section, the vast majority of small hydropower station were demolished. And all the houses in the reserve were demolished. The roads in the reserve have been converted to forestland. In 2017, regional landscape ecological pattern had a series of significant changes. Meanwhile, our experiment followed the event closely, which could prove the robustness of aforesaid research. Although the experimental time had some differences, it also followed the event closely.

  1. The authors refer to five analyzes and three methods and comment that they are applied respectively, but it is not clear which method was used for which analysis (5 and 3 do not correspond). Correct wording.

Thanks so much for your careful reading and probing. In lines 215-218, we has corrected the sentences. The analysis of driving forces of landscape patterns was based on Logistic Regression Model. The simulation of habitat quality was based on CA-Markov Model. The verification of simulation results was based on Participatory Scenario Method.

  1. The authors do not described The Participatory Scenario Method used. Describe it.

Thank you for pointing this out. The Participatory Scenario Method has described in lines 277-281.

  1. The layout of Table 1 is not clear at the bottom left.And there are some visualization problems.

We appreciate these valuable comments. In Table 1, the codes have been included in notes to the table. And we have added the same zoom views for both so that to improve the level of visualization.

Reviewer 2 Report

The main problem with this paper is the quality of the English. Unless the English and the organization are improved, it is difficult to assess the paper’s quality. I know it is unfair for non-native English speakers to write and try to publish the work in English (I’m myself a non-native English speaker), but at the moment this paper needs a huge revision. 

Notice that there are also some problems with the organization of the paper; see examples below. 

Concerning the technical aspects, I couldn’t find any major problems but, as I said, the poor English and organization of the paper may (inadvertently) hide some technical problems. 

It is also a rather long paper, maybe it can be divided into two. For instance, the two case studies presented at the end may be another paper.

Details:

-       A few examples of poorly written sentences:

Lines 71:72, “These measures still do not completely stop farmers from relying on natural resources. 

Lines 612:613, “Free electricity policy cancels to cause farmer to have psychological loss

Lines 635:636, “…cannot be completely avoided in the current situation of contradiction (as shown 635 in Figure 7)

Lines 650:651, “The paper then consulted farmers for countermeasures un- 650 der different scenarios

Lines 658:659, “The paper 658 consults with farmers…

Lines 704:705, “In order to 704 avoid repeated construction caused by system shock…

-       Some statements should be followed by a reference, e.g., 

Lines 560:561 “With the passage of time, farmers' dependence on resources 560 increases exponentially ” 

-       In the logistic model, what does the variable “Road” stand for? The total length of the roads in the study area?

-       “CA” is only defined in line 225, but used extensively before.

-       Some abbreviations are never defined, e.g. “ROC”.

-       Use of abbreviations in the main text defined in the tables (and even there not in a clear way). For example, 

o   Line 545: “HH species” or 

o   Line 550: “LH students”

-       The “notes” shown after the figure captions, should be part of the captions.

-       Some figures do not have scale (fig. 7 and 9), others use “miles”.  I would feel more comfortable with km, but that depends on the journal rules.

Author Response

We are grateful to the editor and reviewers for their positive and constructive comments and criticisms concerning our manuscript, Quantifying the Evolution of Giant Panda Habitats in Sichuan Province under Different Scenarios(diversity-1912699). These comments and criticisms are very helpful for revising and improving our paper. We have made necessary corrections and changes in response to them. Below, our concrete response (indented) follows each specific comment (italicized) of the reviewers. We hope that you will find our revised manuscript is acceptable for publication. Of course, we will make additionally changes if there remain unaddressed or inadequately addressed comments.

Reviewer #2:

  1. The main problem with this paper is the quality of the English. Unless the English and the organization are improved, it is difficult to assess the paper’s quality. I know it is unfair for non-native English speakers to write and try to publish the work in English (I’m myself a non-native English speaker), but at the moment this paper needs a huge revision.

Thanks for your observation and suggestion. We have improved the quality of the English with the help of native English speakers. And we have revised these sentences in the manuscript. In lines 69-70, the sentence has revised as “These measures have not stopped farmers from relying on natural resources, and the root of this phenomenon has been identified as the inertia of farmers.”. In lines 647-648, the sentence has revised as “The cancelation of the free electricity policy cancels caused farmers to experience a psychological loss.”. In line 671, the sentence has revised as “cannot be completely avoided in the current contradictory situation”. In lines 687-688, the sentence has revised as “The researchers then consulted farmers about countermeasures under different scenarios”. In lines 695-698, the sentence has revised as “The researchers consulted farmers about the countermeasures in different scenarios and constructed strategies for four scenarios in Changhe Village according to the clustering principle by sorting farmers' opinions.”. In lines 742-745, the sentence has revised as “To avoid repeated construction caused by system shock, the local government and the reserve administration in the region adhere to the policy of maintaining the existing system, that is, neither increasing nor reducing farmers' resource utilization activities.”. And some statements has added the reference, such as in line 592.

  1. In the logistic model, what does the variable “Road”stand for? The total length of the roads in the study area?

We appreciate these valuable comments. In this paper, “Road” is a class of raster data, which represents the road exist or not. And“Road” is 0-1 variables, which shows the  track of human disturbance.

  1. “CA”is only defined in line 225, but used extensively before. And some abbreviations are never defined.

Thank you for pointing this out. In line 33 and 100, CA has been defined as Cellular Automata. And ROC has been defined in lines 479-480. HH, HL, M, LH and LL have been defined in Table 3.

  1. Some figures do not have scale (fig. 7 and 9), others use “miles”. I would feel more comfortable with km, but that depends on the journal rules.

Thanks so much for your careful reading and probing. Fig. 7 and 9 are hand-drawn maps, which has drawing by participatory experiment. And all the information was provided by the interviewee. The original hand-drawn maps are in the attachment.

Round 2

Reviewer 1 Report

The authors made the suggested corrections to the paper.

Author Response

We are grateful to the editor and reviewers for their positive and constructive comments and criticisms concerning our manuscript, Quantifying the Evolution of Giant Panda Habitats in Sichuan Province under Different Scenarios(diversity-1912699).

Reviewer 2 Report

I commend the authors for the work done improving the English and the presentation overall (in a short period of time). There are still strange expressions, such as:

“precious endangered wildlife conservation“

If you type this expression in Google with inverted commas you won’t find it. I suggest removing the “precious”. 

Another example: The expression “wealth effect” in the following sentence doesn’t seem “natural”:

“Due to the wealth effect, men, middle-aged workers, and 83 young people from the nature reserves and surrounding communities migrate to the cities 84 where they can earn higher incomes ”

The English of the figures should also be revised. For example, in Figure 6 it is written several times “was establish” while it should be “was established”.

But these are just a few examples.

Author Response

We are grateful to the editor and reviewers for their positive and constructive comments and criticisms concerning our manuscript, Quantifying the Evolution of Giant Panda Habitats in Sichuan Province under Different Scenarios(diversity-1912699). These comments and criticisms are very helpful for revising and improving our paper. We have made necessary corrections and changes in response to them. We hope that you will find our revised manuscript is acceptable for publication. Of course, we will make additionally changes if there remain unaddressed or inadequately addressed comments.

Reviewer #2:

  1. “precious endangered wildlife conservation”I suggest removing the “precious”.

Thanks for your observation and suggestion. The “precious” has removed in line 16 and line 45.

  1. The expression “wealth effect”in the following sentence doesn’t seem “natural”.

We appreciate these valuable comments. In this paper, the expression “wealth effect” has been revised as “difference of income” in line 84, which aimed to reflect the urban-rural income gap.

  1. The English of the figures should also be revised. For example, in Figure 6 it is written several times “was establish” while it should be “was established”.

Thank you for pointing this out. The English of the figures and tables have been revised in Figure 6, Figure 8, Table 7 and Table 9, such as “was establish” revise to “was established”.

Round 3

Reviewer 2 Report

My main issue with the paper is the quality of the presentation.   The quality of the English is still rather poor.  Take for example the following two sentences:  

"Participatory Scenario could take the form of workshop. It aimed to explore how to do in the future and how to change. "

The first sentence does not “agree” with the second. One possible interpretation is:   "Participatory Scenario TOOK the form of workshop. It aimed to explore how to do in the future and how to change. "   still “how to do in the future” remains strange: to do what?   Another possibility is:    "Participatory Scenario could take the form of workshop. It WOULD AIM to explore how to do in the future and how to change. "   But this was just a sentence taken from a randomly chosen page.   Unless, the English is properly fixed, I don’t see the point of going back and forward with the manuscript.   

Author Response

We are grateful to the editor and reviewers for their positive and constructive comments and criticisms concerning our manuscript, Quantifying the Evolution of Giant Panda Habitats in Sichuan Province under Different Scenarios(diversity-1912699). These comments and criticisms are very helpful for revising and improving our paper. We have made necessary corrections and changes in response to them. We hope that you will find our revised manuscript is acceptable for publication. Of course, we will make additionally changes if there remain unaddressed or inadequately addressed comments.

Reviewer #2:

  1. The main issue with the paper is the quality of the presentation.

Thanks for your observation and suggestion. We have used English Language Editing Services to help edit the English of the paper. And the article used revision mode. These changes can be easily viewed by the editors and reviewers.

  1. Check the informationin the paper.

We appreciate these valuable comments. In this paper, the information of Funding has been revised as “This research was funded by Ministry of Education, Humanities and Social Sciences Project (Grant number: 22YJCZH150) and National Nature Science Foundation of China (Grant numbers: 71761147003).”

Round 4

Reviewer 2 Report

I congratulate the authors for having tremendously improved the quality of the presentation.